# Incorporating functional priors improves polygenic prediction accuracy in UK Biobank and 23andMe data sets

Carla Márquez-Luna [1,2,3✉], Steven Gazal [2,3], Po-Ru Loh [2,4,5], Samuel S. Kim [2,6], Nicholas Furlotte[7], Adam Auton[7], 23andMe Research Team* & Alkes L. Price [1,2,4✉]

Polygenic risk prediction is a widely investigated topic because of its promising clinical applications. Genetic variants in functional regions of the genome are enriched for complex trait heritability. Here, we introduce a method for polygenic prediction, LDpred-funct, that leverages trait-specific functional priors to increase prediction accuracy. We fit priors using the recently developed baseline-LD model, including coding, conserved, regulatory, and LD-related annotations. We analytically estimate posterior mean causal effect sizes and then use cross-validation to regularize these estimates, improving prediction accuracy for sparse architectures. We applied LDpred-funct to predict 21 highly heritable traits in the UK Biobank (avg $N = 373$ K as training data). LDpred-funct attained a $+4.6\%$ relative improvement in average prediction accuracy (avg prediction $R^2 = 0.144$; highest $R^2 = 0.413$ for height) compared to SBayesR (the best method that does not incorporate functional information). For height, meta-analyzing training data from UK Biobank and 23andMe cohorts ($N = 1107$ K) increased prediction $R^2$ to 0.431. Our results show that incorporating functional priors improves polygenic prediction accuracy, consistent with the functional architecture of complex traits.

[1] Department of Biostatistics, Harvard T.H. Chan School of Public Health, Boston, MA, USA. [2] Program in Medical and Population Genetics, Broad Institute of Harvard and MIT, Cambridge, MA, USA. [3] Charles R. Bronfman Institute for Personalized Medicine, Icahn School of Medicine at Mount Sinai, New York, NY, USA. [4] Department of Epidemiology, Harvard T.H. Chan School of Public Health, Boston, MA, USA. [5] Division of Genetics, Department of Medicine, Brigham and Women's Hospital and Harvard Medical School, Boston, MA, USA. [6] Department of Electrical Engineering and Computer Science, Massachusetts Institute of Technology, Cambridge, MA, USA. [7] 23andMe Inc., Mountain View, CA, USA. *A list of authors and their affiliations appears at the end of the paper. ✉email: cmarquezluna@alumni.harvard.edu; aprice@hsph.harvard.edu

Genetic variants in functional regions of the genome are enriched for complex trait heritability[1–6]. In this study, we aim to leverage functional priors to improve polygenic prediction[7,8]. Several studies have shown that incorporating prior distributions on causal effect sizes can improve prediction accuracy[9–16], compared to standard Best Linear Unbiased Prediction (BLUP) or Pruning + Thresholding methods[17–22]. Recent efforts to incorporate functional information have produced promising results[23,24] (see P + T-funct-LASSO and AnnoPred results in all main figures below), but maybe limited by dichotomizing between functional and non-functional variants[23] or restricting their analyses to genotyped variants[24].

Here, we introduce a method, LDpred-funct, for leveraging trait-specific functional priors to increase polygenic prediction accuracy. We fit functional priors using our recently developed baseline-LD model[25], which includes coding, conserved, regulatory, and LD-related annotations. LDpred-funct first analytically estimates posterior mean causal effect sizes, accounting for functional priors and LD between variants. LDpred-funct then uses cross-validation within validation samples to regularize causal effect size estimates in bins of different magnitude, improving prediction accuracy for sparse architectures. We show that LDpred-funct attains higher polygenic prediction accuracy than other methods in simulations with real genotypes, analyses of 21 highly heritable UK Biobank traits, and meta-analyses of height using training data from UK Biobank and 23andMe cohorts.

## Results

**Simulations**. We performed simulations using real genotypes from the UK Biobank interim release and simulated phenotypes (see Methods). We simulated quantitative phenotypes with SNP-heritability $h_g^2 = 0.5$, using 476,613 imputed SNPs from chromosome 1. We selected either 2000 or 5000 variants to be causal; we refer to these as sparse and polygenic architectures, respectively. We sampled normalized causal effect sizes from normal distributions with variances based on expected causal per-SNP heritabilities under the baseline-LD model[25], fit using stratified LD score regression (S-LDSC)[5,25] applied to height summary statistics from British-ancestry samples from the UK Biobank interim release. We randomly selected 10,000, 20,000, or 50,000 unrelated British-ancestry samples as training samples, and we used 7585 unrelated samples of non-British European ancestry as validation samples. By restricting simulations to chromosome 1 ($\approx$1/10 of SNPs), we can extrapolate results to larger sample sizes ($\approx$10× larger; see Application to 21 UK Biobank traits), analogous to the previous work[16].

We compared prediction accuracies ($R^2$) for seven main methods: P + T[18,19], LDpred[16], SBayesR[9], P + T-funct-LASSO[23], AnnoPred[24], LDpred-funct-inf and LDpred-funct (see Methods). Results are reported in Fig. 1 (main simulations) and Supplementary Fig. 1 (additional values of the number of causal variants); numerical results are reported in Supplementary Tables 1 and 2. Among methods that do not use functional information, the prediction accuracy of LDpred was higher than P + T (particularly for the polygenic architecture), consistent with previous work[8,16] (see Supplementary Tables 3 and 4 for optimal tuning parameters; surprisingly, at $= 50$ K training samples, LDpred is optimized by assuming that 100% of SNPs are causal). SBayesR attained a substantial improvement vs. LDpred at $N = 10$ K training samples (+19% relative improvement for sparse architecture and +8.6% relative improvement for polygenic architecture) but attained prediction $R^2$ close to 0 at larger sample sizes ($N = 20$ K and $N = 50$ K), perhaps because the

algorithm failed to converge (Supplementary Table 1; results not included in Fig. 1).

Incorporating functional information via LDpred-funct-inf (a method that does not model sparsity) produced improvements that varied with sample size (+4.7% relative improvement for sparse architecture and +4.8% relative improvement for polygenic architecture at $N = 50$ K training samples, compared to LDpred; smaller improvements at smaller sample sizes). These results are consistent with the fact that LDpred is known to be sensitive to model assumptions at large sample sizes[16]. Accounting for sparsity using LDpred-funct further improved prediction accuracy, particularly for the sparse architecture (+7.3% relative improvement for sparse architecture and +5.4% relative improvement for polygenic architecture at $N = 50$ K training samples, compared to LDpred; smaller improvements at smaller sample sizes). LDpred-funct attained substantially higher prediction accuracy than P + T-funct-LASSO in most settings (+11% relative improvement for sparse architecture and +18% relative improvement for polygenic architecture at $N = 50$ K training samples; smaller improvements at smaller sample sizes). LDpred-funct also attained higher prediction accuracy than AnnoPred at large sample sizes (+5.7% relative improvement for sparse architecture and +3.7% relative improvement for polygenic architecture at $N = 50$ K training samples; smaller differences at smaller sample sizes) (see Supplementary Table 5 for optimal tuning parameters; surprisingly, at $N = 50$ K training samples, AnnoPred is optimized by assuming that 100% of SNPs are causal, analogous to LDpred). The difference in prediction accuracy between LDpred and each other method, as well as the difference in prediction accuracy between LDpred-funct and each other method, was statistically significant in most cases (see Supplementary Table 2 e.g. vs. AnnoPred: $P < 10^{-125}$ for sparse architecture and $P < 10^{-75}$ for polygenic architecture at $N = 50$ K training samples). Simulations with 1000 or 10,000 causal variants generally recapitulated these findings, although SBayesR, P + T-funct-LASSO and AnnoPred performed better than LDpred-funct for the very sparse architecture at $N = 10$ K (Supplementary Table 1).

The average running time for all 7 methods is reported in Supplementary Table 6. We separately report the time to estimate posterior mean causal effect sizes, and the time to compute LD matrices (not applicable for LDpred-funct-inf and LDpred-funct) (we do not include the time to compute polygenic risk scores, which is small in comparison and depends on the number of validation samples). For the two methods with the highest prediction $R^2$ in analyses of real UK Biobank traits (LDpred-funct and AnnoPred; see below), the average running time was 71 min for LDpred-funct vs. 5249 min for AnnoPred, not including the time to compute LD matrices.

We performed four secondary analyses. First, we assessed the calibration of each method by checking whether regression of true vs. predicted phenotype yielded a slope of 1. We determined that LDpred-funct was well-calibrated (regression slope 0.98–0.99), LDpred and AnnoPred were fairly well-calibrated (regression slope 0.85–1.00), and other methods were not well-calibrated (Supplementary Table 7). Second, we assessed the sensitivity of LDpred-funct to the choice of $K = 40$ posterior mean causal effect size bins to regularize effect sizes in our main simulations. We determined that results were not sensitive to this parameter (Supplementary Table 8); slightly higher values of $K$ performed slightly better, but we did not finely optimize this parameter. Third, we evaluated a cheating version of LDpred-funct that utilized the true baseline-LD model parameters used to simulate the data, instead of estimating these parameters from the data (LDpred-funct-cheat). LDpred-funct-cheat performed only slightly better than LDpred-funct, indicating that LDpred-funct is

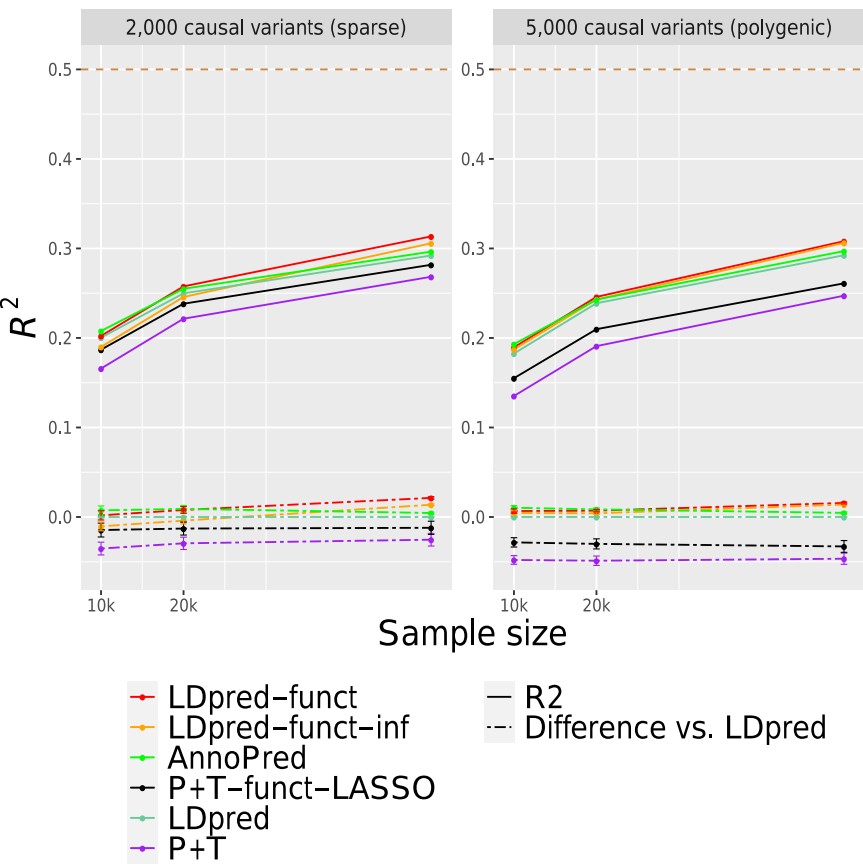

**Fig. 1 Accuracy of 6 polygenic prediction methods in simulations using UK Biobank genotypes.** We report results for P + T, LDpred, P + T-funct-LASSO, AnnoPred, LDpred-funct-inf, and LDpred-funct in chromosome 1 simulations with 2000 causal variants (sparse architecture) and 5000 causal variants (polygenic architecture). Results are presented mean $R^2$ values averaged across 100 simulations. Bottom dashed lines denote differences vs. LDpred; error bars represent 95% confidence intervals. The top dashed line denotes simulated SNP-heritability of 0.5. Results for other values of the number of causal variants are reported in Supplementary Fig. 1, and numerical results are reported in Supplementary Tables 1 and 2. Source data are provided as a Source Data file.

not sensitive to the imperfect estimation of functional enrichment parameters (see Supplementary Table 9). Fourth, we simulated traits with lower SNP-heritability ($h_g^2 = 0.25$) (see Supplementary Table 10). We determined that the improvements attained by LDpred-funct were smaller in these simulations (e.g. +6.9% relative improvement vs. AnnoPred and −1.0% relative improvement vs. LDpred for sparse architecture, +3.4% improvement vs. AnnoPred and +0.6% relative improvement vs. LDpred for polygenic architecture at $N = 50$ K training samples; smaller improvements at smaller sample sizes).

**Application to 21 UK Biobank traits**. We applied P + T, LDpred, SBayesR, P + T-funct-LASSO, AnnoPred, LDpred-funct-inf, and LDpred-funct to 21 UK Biobank traits (14 quantitative traits and 7 binary traits; Supplementary Tables 11 and 12). We analyzed training samples of British-ancestry (avg $N = 373$ K) and validation samples of non-British European ancestry (avg $N = 22$ K). We included 6,334,603 imputed SNPs in our analyses (see Methods). We computed summary statistics and $h_g^2$ estimates from training samples using BOLT-LMM v2.3[26] (see Supplementary Table 13). We estimated trait-specific functional enrichment parameters for the baseline-LD model[25] by running S-LDSC[5,25] on these summary statistics. Results for quantitative traits are reported in Fig. 2 and Supplementary Table 14, and results for binary traits are reported in Fig. 3 and Supplementary Table 15. Differences between each main prediction method and

either LDpred or LDpred-funct (and block-jackknife standard errors on these differences) are reported in Supplementary Table 16, and averages across all 21 traits for main and secondary prediction methods are reported in Supplementary Table 17.

Among methods that do not use functional information, LDpred outperformed P + T (+18% relative improvement in average prediction $R^2$), consistent with simulations under a polygenic architecture (see Supplementary Tables 18 and 19 for optimal tuning parameters) and with the previous work[8,16]. LDpred also outperformed LDpred-inf, a method that does not model sparsity (see Supplementary Table 17). The exclusion of long-range LD regions (see Methods) was critical to LDpred performance, as running LDpred without excluding long-range LD regions (as implemented in a previous version of this paper[27]) performed much worse (see Supplementary Table 17). SBayesR outperformed LDpred (+5.3% relative improvement in average prediction $R^2$), with no convergence issues in the full UK Biobank analysis (but see below for 113 K interim UK Biobank analysis); we note that expanding the set of SNPs analyzed worsened the performance of SBayesR (see below).

Incorporating functional information via LDpred-funct-inf (a method that does not model sparsity) performed only slightly better than LDpred (+0.9% improvement in average prediction $R^2$), but greatly outperformed LDpred-inf (+19% relative improvement, $P < 10^{-20}$ for the difference using two-sided $z$-test based on the block-jackknife standard error in Supplementary Table 20). Accounting for sparsity using LDpred-funct

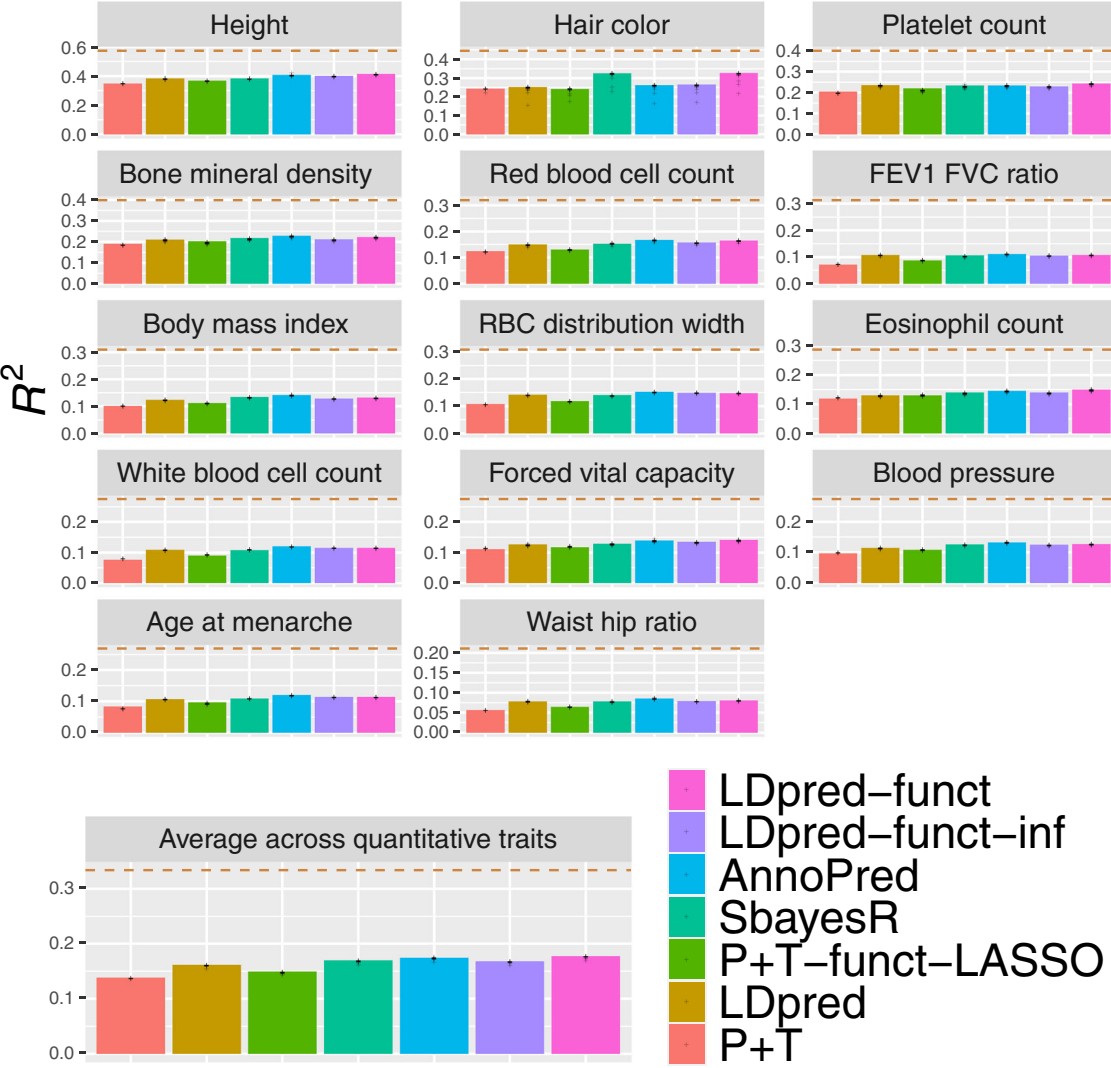

**Fig. 2 Accuracy of 7 polygenic prediction methods across 14 UK Biobank quantitative traits.** We report results for P + T, LDpred, SBayesR, P + T-funct-LASSO, AnnoPred, LDpred-funct-inf and LDpred-funct. Dashed lines denote estimates of SNP-heritability. Numerical results are reported in Supplementary Table 14. Abbreviations: Red Blood Cell Distribution Width (RBD distribution width), forced expiratory volume in one second (FEV1), and forced vital capacity (FVC). Data points represent the prediction accuracy values obtained via block-jacknife over 200 genomic blocks. Source data are provided as a Source Data file.

substantially improved prediction accuracy (+10%, +4.6%, +7.4% relative improvements in average prediction $R^2$ vs. LDpred, SBayesR, LDpred-funct-inf; $P < 2 \times 10^{-4}$, $P = 0.04$, $P < 2 \times 10^{-4}$ for differences using two-sided $z$-test based on the block-jackknife standard error in Supplementary Table 16; average prediction $R^2 = 0.144$; highest $R^2 = 0.413$ for height), consistent with simulations. The relative improvement in avg prediction $R^2$ for LDpred-funct vs. LDpred was +9.7% for quantitative traits (higher prediction $R^2$ for 14/14 traits), and +11% for binary traits (higher prediction $R^2$ for 5/7 traits). We observed a positive but non-significant correlation across traits between $h_g^2$ and relative improvement (Supplementary Fig. 2), perhaps due to the limited number of data points and/or contribution of other factors (e.g. polygenicity). LDpred-funct also performed substantially better than P + T-funct-LASSO (+20% relative improvement in avg. prediction $R^2$), consistent with simulations under a polygenic architecture. AnnoPred performed slightly but non-significantly worse than LDpred-funct (−2.7% relative change in average prediction $R^2$ for AnnoPred vs. LDpred-funct, $P = 0.35$ for the difference using

two-sided $z$-test based on the block-jackknife standard error in Supplementary Table 16; see Supplementary Table 21 for optimal tuning parameters).

In the above experiments, LDpred-funct analyzed 373 K training samples and 22 K validation samples and used 90% of the validation samples to estimate regularization weights (and the remaining validation samples to compute predictions) in each cross-validation fold. It is possible that incorporating data from an additional 20 K samples could confer an unfair advantage for LDpred-funct compared to other methods. To assess this, we performed three additional experiments. First, we repeated the LDpred-funct analyses using smaller validation sample sizes (as low as 1 K), again using 10-fold cross-validation. We determined that results were little changed (Supplementary Table 22). Second, we repeated the LDpred-funct analyses using only 1 K of the 22 K validation samples to estimate regularization weights and the remaining validation samples to compute predictions. Again, we determined that results were little changed (Supplementary Table 23). As the use of 1 K samples to estimate validation weights is a trivial number of additional samples compared to

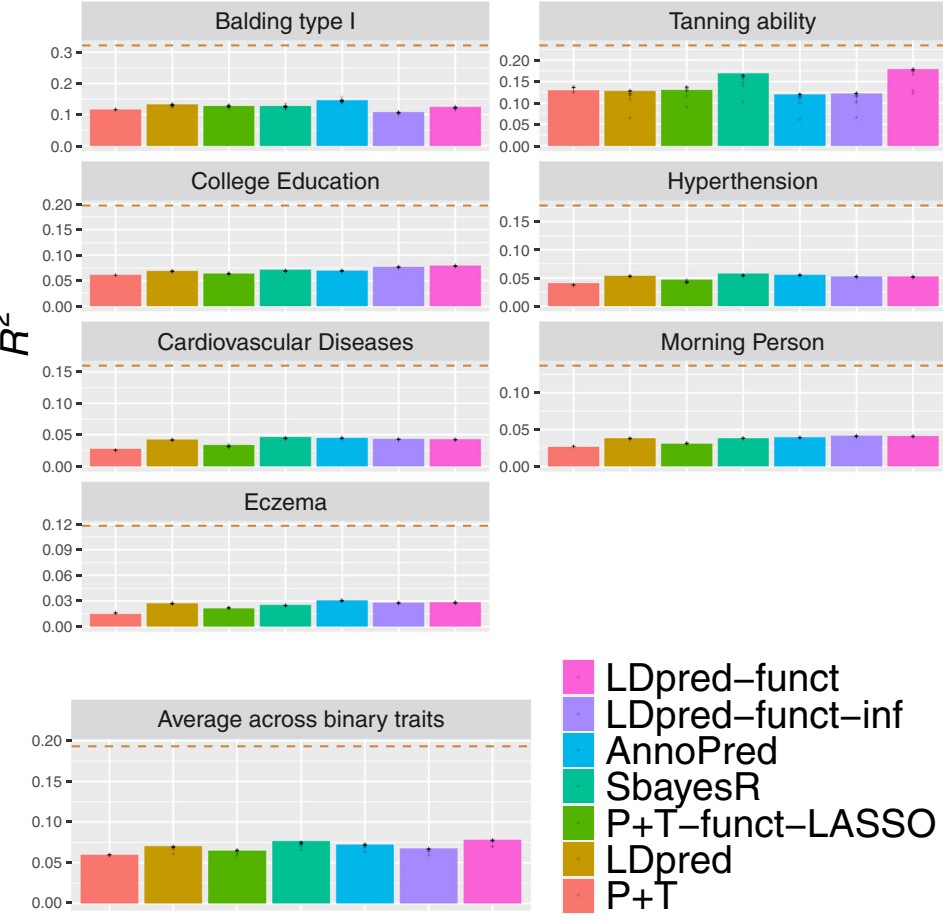

**Fig. 3 Accuracy of 7 polygenic prediction methods across 7 UK Biobank binary traits.** We report results for P + T, LDpred, SBayesR, P + T-funct-LASSO, AnnoPred, LDpred-funct-inf and LDpred-funct. Dashed lines denote estimates of SNP-heritability. Numerical results are reported in Supplementary Table 15. Data points represent the prediction accuracy values obtained via block-jacknife over 200 genomic blocks. Source data are provided as a Source Data file.

373 K training samples. Third, we repeated the analysis using 5 K samples omitted from the set of training samples to estimate regularization weights (we recomputed BOLT-LMM association statistics using the reduced set of 404 K training samples) and the full set of 22 K validation samples to compute predictions. Again, we determined that results were little changed (Supplementary Table 24). We conclude from these experiments that LDpred-funct does not owe its advantage to incorporating data from a substantial number of additional samples.

We performed 13 secondary analyses. First, we assessed the calibration of each method by checking whether regression of true vs. predicted phenotype yielded a slope of 1. As in our simulations, we determined that LDpred-funct was well-calibrated (average regression slope: 0.98), LDpred and AnnoPred were fairly well-calibrated (average regression slope: 0.89 and 0.83, respectively), and other methods were not well-calibrated (Supplementary Table 25). Second, we assessed the sensitivity of LDpred-funct to the average value of $K = 58$ posterior mean causal effect size bins to regularize effect sizes in these analyses (see Eq. (6) and Supplementary Table 13). We determined that results were not sensitive to the number of bins (Supplementary Table 26). Third, we determined that functional enrichment information is far less useful when restricting to genotyped variants (e.g. −6.9% relative change in avg prediction $R^2$ for LDpred-funct vs. LDpred when both methods are restricted to typed variants; Supplementary Table 17), likely because tagging variants may not belong to enriched functional annotations.

Fourth, we repeated the SBayesR analysis using the 2.9 M SNP set instead of the 1.1 M SNP set (see Methods), but determined that this substantially worsened the performance of SBayesR (Supplementary Table 17). Fifth, we evaluated a modification of P + T-funct-LASSO in which different weights were allowed for the two predictors (P + T-funct-LASSO-weighted; see Methods), but results were little changed (+1.1% relative improvement in avg prediction $R^2$ vs. P + T-funct-LASSO; Supplementary Table 17). Sixth, we obtained similar results for P + T-funct-LASSO when defining the "high-prior" (HP) SNP set using the top 5% of SNPs with the highest per-SNP heritability, instead of the top 10% (see Supplementary Table 17). Seventh, we determined that incorporating baseline-LD model functional enrichments that were meta-analyzed across traits (31 traits from ref. [25]), instead of the trait-specific functional enrichments used in our primary analyses, slightly reduced the prediction accuracy of LDpred-funct-inf (Supplementary Table 17). Eighth, to assess whether the improvement of LDpred-funct is specific to the 75 functional annotations of the baseline-LD model, we implemented an analogous method that uses 75 random annotations (LDpred-funct (random)). We determined that LDpred-funct attained a 13% relative improvement in average prediction $R^2$ vs. LDpred-funct (random), which performed similarly to LDpred (3.1% decrease in average prediction $R^2$ vs. LDpred) (Supplementary Table 17). This implies that the improvement of LDpred-funct is specific to the 75 functional annotations of the baseline-LD model. We further note that a method that does not use

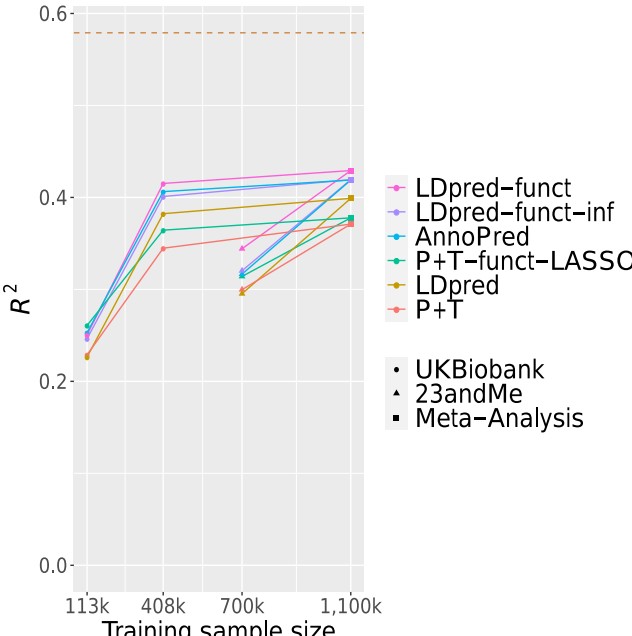

**Fig. 4 Accuracy of 6 prediction methods in height meta-analysis of UK Biobank and 23andMe cohorts.** We report results for P + T, LDpred, P + T-funct-LASSO, AnnoPred, LDpred-funct-inf, and LDpred-funct, for each of 4 training data sets: UK Biobank interim release (113,660 training samples), UK Biobank (408,092 training samples), 23andMe (698,430 training samples) and meta-analysis of UK Biobank and 23andMe (1,107,430 training samples). Nested training data sets are connected by solid lines (e.g. UK Biobank (408 k) and 23andMe are both connected to Meta-Analysis, but not to each other). The dashed line denotes the estimate of SNP-heritability in the UK Biobank. Numerical results are reported in Supplementary Table 27. Block-Jackknife standard errors over 200 genomic jackknife blocks were < 0.028 across each method. Source data are provided as a Source Data file.

functional priors but applies the regularization step of LDpred-funct on top of LDpred-inf (LDpred-inf + sparsity) performed similarly to LDpred-funct (random) (Supplementary Table 17). Ninth, we determined that using our previous baseline model[5], instead of the baseline-LD model[25], slightly reduced the prediction accuracy of LDpred-funct-inf and LDpred-funct (Supplementary Table 17). Tenth, we implemented a method analogous to LDpred-funct that uses functional annotations to restrict to the same set of SNPs with expected per-SNP heritability $\sigma_i^2 > 0$ (2981,166-4,306,498 SNPs depending on the trait; see Methods) but then imposes a constant prior on causal effect sizes (LDpred-funct (constant prior)). We determined that LDpred-funct attained a 4.3% relative improvement in average prediction $R^2$ vs. LDpred-funct (constant prior) (Supplementary Table 17), implying that including a prior informed by functional annotations is better than not including a prior informed by functional annotations. In addition, LDpred-funct (constant prior) attained a 5.5% relative improvement in average prediction $R^2$ vs. LDpred and a 23% relative improvement in average prediction $R^2$ vs. LDpred-inf (Supplementary Table 17), confirming that regularizing causal effect size estimates in bins of different magnitude increases prediction accuracy (also see the comparison of LDpred-funct vs. LDpred-funct-inf above); in addition, some of the improvement of LDpred-funct derives from the removal of (relatively) uninformative SNPs (10% relative improvement for LDpred-funct-inf (constant prior) vs. LDpred-inf; Supplementary Table 17). Eleventh, we determined that inferring functional enrichments using only the SNPs that passed

QC filters and were used for prediction had no impact on the prediction accuracy of LDpred-funct-inf (Supplementary Table 17). Twelveth, we determined that using UK10K (instead of 1000 Genomes) as the LD reference panel had virtually no impact on prediction accuracy (Supplementary Table 17). Thirteenth, we determined that using UK10K (instead of 1000 Genomes) as the LD reference panel had virtually no impact on prediction accuracy (Supplementary Table 17).

**Application to height in meta-analysis of UK Biobank and 23andMe cohorts.** We applied P + T, LDpred-inf, SBayesR, P + T-funct-LASSO, AnnoPred, LDpred-funct-inf, and LDpred-funct to predict height in a meta-analysis of UK Biobank and 23andMe cohorts (see Methods). Training sample sizes were equal to 408,092 for UK Biobank and 698,430 for 23andMe, for a total of 1,106,522 training samples. For comparison purposes, we also computed predictions using the UK Biobank and 23andMe training data sets individually, as well as a training data set consisting of 113,660 British-ancestry samples from the UK Biobank interim release. (The analysis using the 408,092 UK Biobank training samples was nearly identical to the analysis of Fig. 2, except that we used a different set of 5,957,935 SNPs, for consistency throughout this set of comparisons; see Methods.) We used 24,351 UK Biobank samples of non-British European ancestry as validation samples in all analyses.

Results are reported in Fig. 4 and Supplementary Table 27. The relative improvements attained by LDpred-funct-inf and LDpred-funct were broadly similar across all four training data sets (also see Fig. 2), implying that these improvements are not specific to the UK Biobank data set. Interestingly, compared to the full UK Biobank training data set ($R^2 = 0.415$ for LDpred-funct; slightly different from $R^2 = 0.413$ in Fig. 2 due to slightly different SNP set), prediction accuracies were only slightly higher for the meta-analysis training data set ($R^2 = 0.431$ for LDpred-funct), and were lower for the 23andMe training data set ($R^2 = 0.344$ for LDpred-funct), consistent with the ≈30% higher heritability in UK Biobank as compared to 23andMe and other large cohorts[25,26,28]; the higher heritability in UK Biobank could potentially be explained by lower environmental heterogeneity. We note that in the meta-analysis, we optimized the meta-analysis weights using validation data (similar to ref. [29]), instead of performing a fixed-effect meta-analysis. This approach accounts for differences in heritability as well as sample size, and attained a + 3.3% relative improvement in prediction $R^2$ compared to fixed-effects meta-analysis (see Supplementary Table 27). We note that SBayesR performed similarly to LDpred in height analyses with ≥408 K training samples ($-10\%$ to $+0.2\%$ change in average prediction $R^2$) but attained prediction $R^2$ close to 0 in the height analysis with 113 K training samples, perhaps because the algorithm failed to converge (Supplementary Table 27; results not included in Fig. 4).

**Discussion**

We have shown that leveraging trait-specific functional enrichments inferred by S-LDSC with the baseline-LD model[25] substantially improves polygenic prediction accuracy. Across 21 UK Biobank traits, we attained substantial improvements in average prediction $R^2$ using a method that leverages functional enrichment and performs an additional regularization step to account for sparsity (LDpred-funct). LDpred-funct attained $+10\%$ ($P < 2 \times 10^{-4}$) and $+4.6\%$ ($P = 0.04$) relative improvements compared to LDpred[16] and SBayesR[9], two state-of-the-art methods that do not model functional enrichment. Thus incorporating functional annotations improves polygenic prediction accuracy. We note that our main analyses used baseline-LD model v1.1, but using

the updated baseline-LD model v2.1 yields slightly higher prediction $R^2$ for LDpred-funct-inf and LDpred-funct (Supplementary Table 17).

Two previous studies have highlighted the potential advantages of leveraging functional enrichment to improve prediction accuracy[23,24]. We included both of these methods in all of our analyses. First, ref. [23] introduced a method (which we call P + T-funct-LASSO) that corrects marginal effect sizes for winner's curse using LASSO and incorporates functional data to define high-prior and low-prior SNP sets. LDpred-funct attained a +19% average relative improvement vs. P + T-funct-LASSO across 21 UK Biobank traits. Second, ref. [24] introduced AnnoPred, which uses a Bayesian framework to incorporate functional annotations. AnnoPred models sparsity differently than LDpred-funct, as it uses a point-normal prior to estimating posterior mean effect sizes via Markov Chain Monte Carlo (MCMC), whereas LDpred-funct performs a regularization step to account for sparsity. We note that ref. [24] considered only genotyped variants and binary annotations. As noted above, functional enrichment information is far less useful when restricting to genotyped variants (Supplementary Table 17), likely because tagging variants may not belong to enriched functional annotations; thus, the utility of AnnoPred in more general settings is currently unknown. Here, we determined that AnnoPred performed slightly but non-significantly worse than LDpred-funct($-2.3$% relative change in average prediction $R^2$; $P = 0.35$ for difference) across 21 UK Biobank traits, consistent with slightly worse results for AnnoPred in simulations at large sample sizes. We emphasize that our work combines binary and continuous-valued functional annotations to improve polygenic risk prediction using imputed variants.

Our work has several limitations. First, LDpred-funct analyzes summary statistic training data (which are publicly available for a broad set of diseases and traits[30]), but methods that use raw genotypes/phenotypes as training data have the potential to attain higher accuracy[26]; incorporating functional enrichment information into prediction methods that use raw genotypes/phenotypes as training data remains a direction for future research. Second, the regularization step employed by LDpred-funct to account for sparsity relies on heuristic cross-validation instead of inferring posterior mean causal effect sizes under a prior sparse functional model; we made this choice because the appropriate choice of sparse functional model is unclear, and because inference of posterior means via MCMC may be subject to convergence issues. As a consequence, the improvement of LDpred-funct over LDpred-funct-inf may be contingent on the number of validation samples available for cross-validation; in particular, for very small validation samples, the number of cross-validation bins is equal to 1 (Eq. (6)) and LDpred-funct is identical to LDpred-funct-inf. However, we determined that results of LDpred-funct were little changed when restricting to smaller validation sample sizes (as low as 1000; see Supplementary Table 22) or using all 22 K validation samples but using only 1 K samples to estimate validation weights ((Supplementary Table 23); this implies that LDpred-funct does not owe its advantage to incorporating data from a substantial number of additional samples. Third, we have considered only single-trait analyses, but leveraging genetic correlations among traits has considerable potential to improve prediction accuracy[31,32]. Fourth, we have not considered how to leverage functional enrichment for polygenic prediction in related individuals[33]. Fifth, we have not thoroughly investigated the application LDpred-funct to polygenic prediction in diverse populations[29,34–36] (for which very similar functional enrichments have been reported[37,38]), as our simulations focused exclusively on prediction in Europeans. However, we evaluated the performance of LDpred-funct in predicting 21 UK Biobank traits in diverse populations using European training data (as in recent studies[34,35]). The results were promising, particularly in Africans (+23% vs. LDpred ($P < 10^{-5}$), +18% vs. SBayesR

($P = 0.001$); see Supplementary Table 28), for which distinguishing causal vs. non-causal variants is particularly important due to differences in LD vs. Europeans[39]. A more thorough investigation, e.g. incorporating non-European training data[29], is an important direction for future research. Sixth, we have not performed a comprehensive assessment of how much different functional annotation models contribute to improvements in prediction accuracy, which remains an important future direction, particularly as functional annotation models will improve as increasingly rich functional data is generated. Specifically, the improvements in prediction accuracy that we reported are a function of the baseline-LD model[25], but there are many possible ways to improve this model, e.g. by incorporating tissue-specific enrichments[1–6,40–43], modeling MAF-dependent architectures[44–46], and/or employing alternative approaches to modeling LD-dependent effects[47]; we anticipate that future improvements to the baseline-LD model will yield even larger improvements in prediction accuracy. As an initial step to explore alternative approaches to modeling LD-dependent effects, we repeated our analyses using the baseline-LD + LDAK model (introduced in ref. [48]), which consists of the baseline-LD model plus one additional continuous annotation constructed using LDAK weights[47]. (Recent work has shown that incorporating LDAK weights increases polygenic prediction accuracy in analyses that do not include the baseline-LD model[49].) We determined that results were virtually unchanged (avg prediction $R^2 = 0.1350$ for baseline-LD + LDAK vs. 0.1354 for baseline-LD using LDpred-funct-inf with UK10K SNPs; see Supplementary Tables 17 and 29). Despite these limitations and open directions for future research, our work demonstrates that leveraging functional enrichment using the baseline-LD model substantially improves polygenic prediction accuracy.

## Methods

**Polygenic prediction methods**. We compared 7 main prediction methods: Pruning + Thresholding[18,19] (P + T), LDpred[16], SBayesR[9], P + T with functionally informed LASSO shrinkage[23] (P + T-funct-LASSO), AnnoPred[24], our LDpred-funct-inf method, and our LDpred-funct method; we also included LDpred-inf[16], which is known to attain lower prediction accuracy than LDpred[16], in some of our secondary analyses. P + T, LDpred-inf, LDpred, and SBayesR are polygenic prediction methods that do not use functional annotations; we did not include RSS[12] and SBLUP[11] methods in our comparisons, because ref. [9] reported that SBayesR performed as well or better than both RSS and SBLUP and was more computationally efficient (Fig. 2 and Supplementary Fig. 18 of ref. [9]). P + T-funct-LASSO is a modification of P + T that corrects marginal effect sizes for winner's curse, accounting for functional annotations. AnnoPred is which uses a Bayesian framework to incorporate functional annotations. LDpred-funct-inf is an improvement of LDpred-inf that incorporates functionally informed priors on causal effect sizes. LDpred-funct is an improvement of LDpred-funct-inf that uses cross-validation to regularize posterior mean causal effect size estimates, improving prediction accuracy for sparse architectures. Each method is described in greater detail below. In both simulations and analyses of real traits, we used squared correlation ($R^2$) between predicted phenotype and true phenotype in a held-out set of samples as our primary measure of prediction accuracy.

**P + T**. The P + T method builds a polygenic risk score (PRS) using a subset of independent SNPs obtained via informed LD-pruning[19] (also known as LD-clumping) followed by P-value thresholding[18]. Specifically, the method has two parameters, $R^2_{LD}$ and $P_T$, and proceeds as follows. First, the method prunes SNPs based on a pairwise threshold $R^2_{LD}$, removing the less significant SNP in each pair. Second, the method restricts to SNPs with an association P-value below the significance threshold $P_T$. Letting $M$ be the number of SNPs remaining after LD-clumping, polygenic risk scores (PRS) are computed as

$$\text{PRS}(P_T) = \sum_{i=1}^{M} \mathbb{1}_{\{P_i < P_T\}} \tilde{\beta}_i g_i, \tag{1}$$

where $\tilde{\beta}_i$ are normalized marginal effect size estimates and $g_i$ is a vector of normalized genotypes for SNP $i$. The parameters $R^2_{LD}$ and $P_T$ are commonly tuned using validation data to optimize prediction accuracy[18,19]. While in theory, this procedure is susceptible to overfitting, in practice, validation sample sizes are typically large, and $R^2_{LD}$ and $P_T$ are selected from a small discrete set of parameter choices, so that overfitting is considered to have a negligible effect[7,18,19,29]. Accordingly, in this work, we consider $R^2_{LD} \in \{0.1, 0.2, 0.5, 0.8\}$ and $P_T \in \{1, 0.3,$

0.1, 0.03, 0.01, 0.003, 0.001, $3 * 10^{-4}, 10^{-4}, 3 * 10^{-5}, 10^{-5}, 10^{-6}, 10^{-7}, 10^{-8}$}, and we always report results corresponding to the best choices of these parameters. The P + T method is implemented in the PLINK software (see Code availability).

**LDpred-inf.** The LDpred-inf method estimates posterior mean causal effect sizes under an infinitesimal model, accounting for LD[16]. The infinitesimal model assumes that normalized causal effect sizes have prior distribution $\beta_i \sim N(0, \sigma^2)$, where $\sigma^2 = h_g^2/M$, $h_g^2$ is the SNP-heritability, and $M$ is the number of SNPs. The posterior mean causal effect sizes are

$$E(\boldsymbol{\beta}|\tilde{\boldsymbol{\beta}}, \mathbf{D}) = \left(\frac{N}{1 - h_l^2} * \mathbf{D} + \frac{1}{\sigma^2}\mathbf{I}\right)^{-1} N * \tilde{\boldsymbol{\beta}}, \quad (2)$$

where $\mathbf{D}$ is the LD matrix between markers, $\mathbf{I}$ is the identity matrix, $N$ is the training sample size, $\tilde{\boldsymbol{\beta}}$ is the vector of marginal association statistics, and $h_l^2 \approx kh_g^2/M$ is the heritability of the $k$ SNPs in the region of LD; following ref. [16] we use the approximation $1 - h_l^2 \approx 1$, which is appropriate when $M >> k$. $\mathbf{D}$ is typically estimated using validation data, restricting to non-overlapping LD windows. We used the default LD window size, which is M/3000. $h_g^2$ can be estimated from raw genotype/phenotype data[26,28] (the approach that we use here; see below), or can be estimated from summary statistics using the aggregate estimator as described in ref. [16]. To approximate the normalized marginal effect size ref. [16] uses the p-values to obtain absolute Z scores and then multiplies absolute Z scores by the sign of the estimated effect size. When sample sizes are very large, p-values may be rounded to zero, in which case we approximate normalized marginal effect sizes $\hat{\beta}_i$ by $\hat{b}_i \frac{\sqrt{2 * p_i * (1 - p_i)}}{\sqrt{\sigma_Y^2}}$, where $\hat{b}_i$ is the per-allele marginal effect size estimate, $p_i$ is the minor allele frequency of SNP $i$, and $\sigma_Y^2$ is the phenotypic variance in the training data. This applies to all the methods that use normalized effect sizes. Although the published version of LDpred requires a matrix inversion (Eq. (2)), we have implemented a computational speedup that computes the posterior mean causal effect sizes by efficiently solving[50] the system of linear equations $(\frac{1}{\sigma^2}\mathbf{I} + N * \mathbf{D})E(\boldsymbol{\beta}|\tilde{\boldsymbol{\beta}}, \mathbf{D}) = N\tilde{\boldsymbol{\beta}}$.

**LDpred.** The LDpred method is an extension of LDpred-inf that uses a point-normal prior to estimating posterior mean effect sizes via Markov Chain Monte Carlo (MCMC)[16]. It assumes a Gaussian mixture prior: $\beta_i \sim N(0, h_g^2/M * p)$ with probability $p$, and $\beta_i \sim 0$ with probability $1 - p$, where $p$ is the proportion of causal SNPs. The method is optimized by considering different values of $p$ (1E−4, 3E−4, 1E−3, 3E−3, 0.01,0.03,0.1,0.3,1); in the special case where 100% of SNPs are assumed to be causal, LDpred is roughly equivalent to LDpred-inf. We excluded SNPs from long-range LD regions (reported in ref. [51]), as our secondary analyses showed that including these regions were suboptimal, consistent with ref. [9].

**SBayesR.** The SBayesR method infers posterior mean causal effect sizes from GWAS summary statistics and an LD matrix[9]. It assumes a finite mixture of normal distributions to account for sparsity, defined as: $\beta_i \sim N(0, \gamma_c h_g^2)$ with probability $\pi_c$, where $c$ ranges from 1 to $C$, the total number of components in the mixture model. We used as input the recommended parameters from ref. [9], with $C = 4$ mixtures with parameters $\gamma_c = (0, 0.01, 0.1, 1.0)$. The method requires a shrunk LD matrix[12]. The authors of ref. [9] made available shrunk LD matrices estimated from 50,000 randomly selected white British individuals from the UK Biobank[51] for two different SNPs sets. The 1.1M SNP set consists of 1,094,841 variants, constructed by restricting 1,365,446 SNPs from HapMap3[52] to MAF > 0.01 and removing strand ambiguous SNPs and long-range LD regions (as reported in ref. [51]). The 2.9M SNP set consists of 2,865,810 variants, constructed by applying LD-pruning ($R^2 > 0.99$) to a larger set of 8 million variants from the UK Biobank[51] with MAF > 0.01, overlapped with a previous large GWAS[53] and present in 1000 Genomes[54]. We note that we could not scale the SBayesR analysis to the full set of 6,334,603 variants used in other analyses due to computational constraints. We used the 1.1M SNP set in our primary analyses as it achieved the highest average prediction $R^2$ in our real traits analyses (see Results section), but we also considered the 2.9 M SNP set in secondary analyses. For analyses that use BOLT-LMM summary statistics we used $N_{effective}$ as reported in ref. [26].

**P + T-funct-LASSO.** Reference[23] proposed an extension of P + T that corrects the marginal effect sizes of SNPs for winner's curse and incorporates external functional annotation data (P + T-funct-LASSO). The winner's curse correction is performed by applying a LASSO shrinkage to the marginal association statistics of the PRS:

$$\text{PRS}_{\text{LASSO}}(P_T) = \sum_{i=1}^{M} sign(\tilde{\beta}_i)||\tilde{\beta}_i| - \lambda(P_T)|\mathbb{1}_{\{P_i < P_T\}}g_i, \quad (3)$$

where $\lambda(P_T) = \Phi^{-1}(1 - \frac{P_T}{2})sd(\tilde{\beta}_i)$, where $\Phi^{-1}$ is the inverse standard normal CDF. Functional annotations are incorporated via two disjoint SNPs sets, representing "high-prior" SNPs (HP) and "low-prior" SNPs (LP), respectively. We define the HP SNP set for P + T-funct-LASSO as the set of SNPs in the top 10% of expected per-SNP

heritability under the baseline-LD model[25], which includes coding, conserved, regulatory, and LD-related annotations, whose enrichments are jointly estimated using stratified LD score regression[5,25] (see Baseline-LD model annotations section). We also performed secondary analyses using the top 5% (P + T-funct-LASSO-top5%). We define $\text{PRS}_{\text{LASSO,HP}}(P_{HP})$ to be the PRS restricted to the HP SNP set, and $\text{PRS}_{\text{LASSO, LP}}(P_{LP})$ to be the PRS restricted to the LP SNP set, where $P_{HP}$ and $P_{LP}$ are the optimal significance thresholds for the HP and LP SNP sets, respectively. We define $\text{PRS}_{\text{LASSO}}(P_{HP}, P_{LP}) = \text{PRS}_{\text{LASSO,HP}}(P_{HP}) + \text{PRS}_{\text{LASSO,LP}}(P_{LP})$. We also performed secondary analyses where we allow an additional regularization to the two PRS: $\text{PRS}_{\text{LASSO}}(P_{HP}, P_{LP}) = \alpha_1\text{PRS}_{\text{LASSO,HP}}(P_{HP}) + \alpha_2\text{PRS}_{\text{LASSO, LP}}(P_{LP})$; we refer to this method as P + T-funct-LASSO-weighted.

**AnnoPred.** AnnoPred[24] uses a Bayesian framework to incorporate functional priors while accounting for LD, optimizing prediction $R^2$ over different assumed values of the proportion of causal SNPs. Reference[24] proposed two different priors for use with AnnoPred. The first prior assumes the same proportion of causal SNPs but different causal effect size variance across functional annotations, and uses a point-normal prior to estimating posterior mean effect sizes via Markov Chain Monte Carlo (MCMC). In the special case where 100% of SNPs are assumed to be causal, AnnoPred is roughly equivalent to LDpred-funct-inf (see below). The second prior assumes different proportions of causal SNPs but the same causal effect size variance across functional annotations. We only consider the first prior, since the second prior cannot be extended to incorporate continuous-valued annotations from the baseline-LD model. We excluded SNPs from long-range LD regions (as reported in ref. [51]) when running AnnoPred. We used the default LD window size, which is M/3000.

**LDpred-funct-inf.** We modify LDpred-inf to incorporate functionally informed priors on causal effect sizes using the baseline-LD model[25], which includes coding, conserved, regulatory, and LD-related annotations, whose enrichments are jointly estimated using stratified LD score regression[5,25]. Specifically, we assume that normalized causal effect sizes have prior distribution $\beta_i \sim N(0, c * \sigma_i^2)$, where $\sigma_i^2$ is the expected per-SNP heritability under the baseline-LD model (fit using training data only) and $c$ is a normalizing constant such that $\sum_{i=1}^{M} \mathbb{1}_{\{\sigma_i^2 > 0\}}c\sigma_i^2 = h_g^2$; SNPs with $\sigma_i^2 \leq 0$ are removed, which is equivalent to setting $\sigma_i^2 = 0$. The posterior mean causal effect sizes are

$$E[\boldsymbol{\beta}|\tilde{\boldsymbol{\beta}}, \mathbf{D}, \sigma_1^2, \ldots, \sigma_{M_+}^2] = \mathbf{W}^{-1}N * \tilde{\boldsymbol{\beta}} = \left[N * \mathbf{D} + \frac{1}{c}\begin{pmatrix} \frac{1}{\sigma_1^2} & \cdots & 0 \\ \vdots & \ddots & \vdots \\ 0 & \cdots & \frac{1}{\sigma_{M_+}^2} \end{pmatrix}\right]^{-1} N * \tilde{\boldsymbol{\beta}},$$

$$(4)$$

where $M_+$ is the number of SNPs with $\sigma_i^2 > 0$. The posterior mean causal effect sizes are computed by solving the system of linear equations $\mathbf{W}E[\boldsymbol{\beta}|\tilde{\boldsymbol{\beta}}, \mathbf{D}, \sigma_1^2, \ldots, \sigma_M^2] = N * \tilde{\boldsymbol{\beta}}$. $h_g^2$ is estimated as described above (see LDpred-inf). $\mathbf{D}$ is estimated using validation data, restricting to windows of size $0.15\%M_+$. In principle, it is possible to use banding to define the LD matrices, where LD between distant pairs of SNPs (10 Mb or more) is rounded to zero[55], but we elected to use the simpler window-based approach (as in ref. [16]).

**LDpred-funct.** We modify LDpred-funct-inf to regularize posterior mean causal effect sizes using cross-validation. We rank the SNPs by their (absolute) posterior mean causal effect sizes, partition the SNPs into $K$ bins (analogous to ref. [56]) where each bin has roughly the same sum of squared posterior mean effect sizes, and determine the relative weights of each bin based on the predictive value in the validation data. Intuitively if a bin is dominated by non-causal SNPs, the inferred relative weight will be lower than for a bin with a high proportion of causal SNPs. This non-parametric shrinkage approach can optimize prediction accuracy regardless of the genetic architecture. In detail, let $S = \sum_i E[\beta_i|\tilde{\beta}_i]^2$. To define each bin, we first rank the posterior mean effect sizes based on their squared values $E[\beta_i|\tilde{\beta}_i]^2$. We define bin $b_1$ as the smallest set of top SNPs with $\sum_{i \in b_1} E[\beta_i|\tilde{\beta}_i]^2 \geq \frac{S}{K}$, and iteratively define bin $b_k$ as the smallest set of additional top SNPs with $\sum_{i \in b_1, \ldots, b_k} E[\beta_i|\tilde{\beta}_i]^2 \geq \frac{kS}{K}$. Let $\text{PRS}(k) = \sum_{i \in b_k} E[\beta_i|\tilde{\beta}_i]g_i$. We define

$$\text{PRS}_{\text{LDpred−funct}} = \sum_{k=1}^{K} \alpha_k \text{PRS}(k), \quad (5)$$

where the bin-specific weights $\alpha_k$ are optimized using validation data via 10-fold cross-validation. For each held-out fold in turn, we split the data so we estimate the weights $\alpha_k$ using the samples from the other nine folds (90% of the validation) and compute PRS on the held-out fold using these weights (10% of the validation); thus, in each cross-validation fold, the validation samples used to estimate regularization weights are disjoint from the validation samples used to compute predictions. We then compute the average prediction $R^2$ across the 10 held-out folds. To avoid overfitting when $K$ is very close to $N$, we set the number of bins ($K$) to be between 1 and 100, such that it is proportional to $h_g^2$ and the number of samples used to

estimate the $K$ weights in each fold is at least 100 times larger than $K$:

$$K = \min\left(100, \left\lceil \frac{0.9N * h_g^2}{100} \right\rceil \right), \qquad (6)$$

where $N$ is the number of validation samples. For highly heritable traits ($h_g^2 \sim 0.5$), LDpred-funct reduces to the LDpred-funct-inf method if there are ~ 200 validation samples or fewer; for less heritable traits ($h_g^2 \sim 0.1$), LDpred-funct reduces to the LDpred-funct-inf method if there are ~ 1000 validation samples or fewer. In simulations, we set $K$ to 40 (based on 7,585 validation samples; see below), approximately concordant with Eq. (6). The value of 100 in the denominator of Eq. (6) was coarsely optimized in simulations, but was not optimized using real trait data. We note that functional annotations are not used in the cross-validation step (although they do impact the posterior mean causal effect size provided as input to this step). Thus, it is likely that SNPs from a given functional annotation will fall into different bins (possibly all of the bins).

Standard errors. Standard errors for the prediction $R^2$ of each method and the difference in prediction $R^2$ between two methods were computed via block-jackknife using 200 genomic jackknife blocks[5]; this is more conservative than computing standard errors based on the number of validation samples, which does not account for variation across a finite number of SNPs. For each method, we first optimized any relevant tuning parameters using the entire genome and then analyzed each jackknife block using those tuning parameters.

**Simulations.** We simulated quantitative phenotypes using real genotypes from the UK Biobank interim release (see below). We used up to 50,000 unrelated British-ancestry samples as training samples, and 7,585 samples of other European ancestries as validation samples (see below). We made these choices to minimize confounding due to shared population stratification or cryptic relatedness between training and validation samples (which, if present, could overstate the prediction accuracy that could be obtained in independent samples[57]), while preserving a large number of training samples. We restricted our simulations to 459,284 imputed SNPs on chromosome 1 (see below), fixed the number of causal SNPs at 2,000 or 5,000 (we also performed secondary simulations with 1000 or 10,000 causal variants), and fixed the SNP-heritability $h_g^2$ at 0.5. We sampled normalized causal effect sizes $\beta_i$ for causal SNPs from a normal distribution with variance equal to $\frac{\sigma_i^2}{p}$, where $p$ is the proportion of causal SNPs and $\sigma_i^2$ is the expected causal per-SNP heritability under the baseline-LD model[25], fit using stratified LD score regression (S-LDSC)[5,25] applied to height summary statistics computed from unrelated British-ancestry samples from the UK Biobank interim release ($N = 113,660$). We computed per-allele effect sizes $b_i$ as $b_i = \frac{\beta_i}{\sqrt{2p_i(1-p_i)}}$, where $p_i$ is the minor allele frequency for SNP $i$ estimated using the validation genotypes. We simulated phenotypes as $Y_j = \sum_i^M b_i g_{ij} + \epsilon_j$, where $\epsilon_j \sim N(0, 1 - h_g^2)$. We set the training sample size to either 10,000, 20,000, or 50,000. The motivation to perform simulations using one chromosome is to be able to extrapolate performance at larger sample sizes[16] according to the ratio $N/M$, where $N$ is the training sample size. We compared each of the seven methods described above. For LDpred-funct-inf and LDpred-funct, for each simulated trait we used S-LDSC (applied to training data only) to estimate baseline-LD model parameters. For LDpred-funct, we report $R^2$ as the average prediction $R^2$ across the 10 held-out folds.

**Full UK Biobank data set.** The full UK Biobank data set includes 459,327 European-ancestry samples and ~ 20 million imputed SNPs[51] (after filtering as in ref. [26], excluding indels and structural variants). We selected 21 UK Biobank traits (14 quantitative traits and 7 binary traits) with phenotyping rate >80% (>80% of females for age at menarche, >80% of males for balding), SNP-heritability $h_g^2 > 0.2$ for quantitative traits, observed-scale SNP-heritability $h_g^2 > 0.1$ for binary traits, and low correlation between traits (as described in ref. [26]). We restricted training samples to 409,728 British-ancestry samples[51], including related individuals (avg $N = 373$ K phenotyped training samples; see Supplementary Table 11 for quantitative traits and Supplementary Table 12 for binary traits). We computed association statistics from training samples using BOLT-LMM v2.3[26]. We have made these association statistics publicly available (see Data availability). We restricted validation samples to 24,436 samples of non-British European ancestry, after removing validation samples that were related ( >0.05) to training samples and/or other validation samples (avg $N = 22$ K phenotyped validation samples; see Supplementary Tables 11 and 12). As in our simulations, we made these choices to minimize confounding due to shared population stratification or cryptic relatedness between training and validation samples (which, if present, could overstate the prediction accuracy that could be obtained in independent samples[57]), while preserving a large number of training samples. We analyzed 6,334,603 genome-wide imputed SNPs, after removing SNPs with minor allele frequency <1%, removing SNPs with imputation accuracy <0.9, and removing A/T and C/G SNPs to eliminate potential strand ambiguity. We used $h_g^2$ estimates from BOLT-LMM v2.3[26] as input to LDpred, AnnoPred, LDpred-funct-inf, and LDpred-funct.

**UK Biobank interim release.** The UK Biobank interim release includes 145,416 European-ancestry samples[58]. We used the UK Biobank interim release both in simulations using real genotypes, and in a subset of analyses of height phenotypes (to investigate how prediction accuracy varies with training sample size).

In our analyses of height phenotypes, we restricted training samples to 113,660 unrelated (≤0.05) British-ancestry samples for which height phenotypes were available. We computed association statistics by adjusting for 10 PCs[59], estimated using FastPCA[60] (see Code availability). For consistency, we used the same set of 24,351 validation samples of non-British European ancestry with height phenotypes as defined above. We analyzed 5,957,957 genome-wide SNPs, after removing SNPs with minor allele frequency <1%, removing SNPs with imputation accuracy <0.9, removing SNPs that were not present in the 23andMe height data set (see below), and removing A/T and C/G SNPs to eliminate potential strand ambiguity.

In our simulations, we restricted training samples to up to 50,000 of the 113,660 unrelated British-ancestry samples, and restricted validation samples to 8441 samples of non-British European ancestry, after removing validation samples that were related ( >0.05) to training samples and/or other validation samples. We restricted the 5,957,957 genome-wide SNPs (see above) to chromosome 1, yielding 459,284 SNPs after QC.

**23andMe height summary statistics.** The 23andMe data set consists of summary statistics computed from 698,430 European-ancestry samples (23andMe customers who consented to participate in research) at 9,898,287 imputed SNPs, after removing SNPs with minor allele frequency <1% and that passed QC filters (which include filters on imputation quality, avg.rsq <0.5 or min.rsq <0.3 in any imputation batch, and imputation batch effects). Analyses were restricted to the set of individuals with >97% European ancestry, as determined via an analysis of local ancestry[61]. Summary association statistics were computed using linear regression adjusting for age, gender, genotyping platform, and the top five principal components to account for residual population structure. The summary association statistics will be made available to qualified researchers (see Data availability).

We analyzed 5,808,258 genome-wide SNPs, after removing SNPs with minor allele frequency <1%, removing SNPs with imputation accuracy <0.9, removing SNPs that were not present in the full UK Biobank data set (see above), and removing A/T and C/G SNPs to eliminate potential strand ambiguity.

**Meta-analysis of full UK Biobank and 23andMe height data sets.** We meta-analyzed height summary statistics from the full UK Biobank and 23andMe data sets. We define

$$\text{PRS}_{\text{meta}} = \gamma_1 \text{PRS}_1 + \gamma_2 \text{PRS}_2, \qquad (7)$$

where $\text{PRS}_i$ is the PRS obtained using training data from cohort $i$. The PRS can be obtained using P + T, P + T-funct-LASSO, LDpred-inf, or LDpred-funct. The meta-analysis weights $\gamma_i$ can either be specified via fixed-effect meta-analysis (e.g. $\gamma_i = \frac{N_i}{\sum_i N_i}$) or optimized using validation data[29]. We use the latter approach, which can improve prediction accuracy (e.g. if the cohorts differ in their heritability as well as their sample size). In our primary analyses, we fit the weights $\gamma_i$ in-sample and report prediction accuracy as adjusted $R^2$ to account for in-sample fitting[29]. We also report results using 10-fold cross-validation: for each held-out fold in turn, we estimate the weights $\gamma_i$ using the other nine folds and compute PRS on the held-out fold using these weights. We then compute the average prediction $R^2$ across the 10 held-out folds.

When using LDpred-funct as the prediction method, we perform the meta-analysis as follows. First, we use LDpred-funct-inf to fit meta-analysis weights $\gamma_i$. Then, we use $\gamma_i$ to compute (meta-analysis) weighted posterior mean causal effect sizes (PMCES) via $\text{PMCES} = \gamma_1 \text{PMCES}_1 + \gamma_2 \text{PMCES}_2$, which are binned into $k$ bins. Then, we estimate bin-specific weights $\alpha_k$ (used to compute (meta-analysis + bin-specific) weighted posterior mean causal effect sizes $\sum_{k=1}^K \alpha_k \text{PMCES}(k)$) using validation data via 10-fold cross-validation.

**Baseline-LD model annotations.** The baseline-LD model (v1.1) contains a broad set of 75 functional annotations (including coding, conserved, regulatory, and LD-related annotations), whose enrichments are jointly estimated using stratified LD score regression[5,25]. For each trait, we used the $\tau_c$ values estimated for that trait to compute $\sigma_i^2$, the expected per-SNP heritability of SNP $i$ under the baseline-LD model, as

$$\sigma_i^2 = \sum_c a_c(i)\tau_c, \qquad (8)$$

where $a_c(i)$ is the value of annotation $c$ at SNP $i$.

Joint effect sizes $\tau_c$ for each annotation $c$ are estimated via

$$E[\chi_i^2] = N \sum_c \tau_c l(i,c) + 1, \qquad (9)$$

where $l(i,c)$ is the LD score of SNP $i$ with respect to annotation $a_c$ and $\chi_i^2$ is the chi-square statistic for SNP $i$. We note that $\tau_c$ quantifies effects that are unique to annotation $c$. In all analyses of real phenotypes, $\tau_c$ and $\sigma_i^2$ were estimated using training samples only.

In our primary analyses, we used 489 unrelated European samples from phase 3 of the 1000 Genomes Project[54] as the reference data set to compute LD scores, as in ref. [25].

To verify that our 1000 Genomes reference data set produces reliable LD estimates, we repeated our LDpred-funct analyses using S-LDSC with 3,567 unrelated individuals from UK10K[62] as the reference data set (as in ref. [48]), ensuring a closer ancestry match with British-ancestry UK Biobank samples. We also repeated our LDpred-funct analyses using S-LDSC with the baseline-LD + LDAK model (instead of the baseline-LD model), with UK10K as the reference data set. The baseline-LD + LDAK model (introduced in ref. [48]) consists of the baseline-LD model plus one additional continuous annotation constructed using LDAK weights[47], which has values $(p_j(1 - p_j))^{1+\alpha} w_j$, where $\alpha = -0.25$, $p_j$ is the allele frequency of SNP $j$, and $w_j$ is the LDAK weight of SNP $j$ computed using UK10K data.

**Reporting summary**. Further information on research design is available in the Nature Research Reporting Summary linked to this article.

## Data availability

Source data are provided with this paper. We used BOLT-LMM v2.3 association statistics: https://data.broadinstitute.org/alkesgroup/UKBB/UKBB_409K/. The baseline-LD annotations (v.2.1) used to compute functional enrichments in the primary analysis are available at https://alkesgroup.broadinstitute.org/LDSCORE/1000G_Phase3_baseline_v1.2_ldscores.tgz. 1000 Genomes Project, http://www.1000genomes.org/. Access to the UK10K data used in the secondary analysis is available via application in https://www.uk10k.org/data_access.html. Access to the UK Biobank resource is available via application in http://www.ukbiobank.ac.uk/. 23andMe height association statistics: The full summary statistics for the 23andMe height GWAS data will have restricted access, and will be made available through 23andMe to qualified researchers under an agreement with 23andMe that protects the privacy of the 23andMe participants. Please visit https://research.23andme.com/collaborate/#publication for more information and to apply to access the data. SBayesR shrunk and sparse LD matrices can be downloaded from Zenodo public repository https://zenodo.org/, for both 1.09 million HapMap3 (https://doi.org/10.5281/zenodo.3350914) and 2.8 million pruned variants (https://doi.org/10.5281/zenodo.3375373). Source data are provided with this paper.

## Code availability

Software implementing the LDpred-funct-inf and LDpred-funct[63]: https://www.hsph.harvard.edu/alkes-price/software (https://doi.org/10.5281/zenodo.4579879). LDscore regression v1.0.1 software: https://github.com/bulik/ldsc. BOLT-LMM v2.3 software http://data.broadinstitute.org/alkesgroup/BOLT-LMM/. FASTPCA is available in EIGENSOFT(7.2.1) at https://github.com/DReichLab/EIG/archive/v7.2.1.tar.gz (more details in https://www.hsph.harvard.edu/alkes-price/software). AnnoPred: https://github.com/yiminghu/AnnoPred. SBayesR 2.0 software: http://cnsgenomics.com/software/gctb/. LDAK version 5 is available at http://dougspeed.com/downloads/. Plink 2.0 is available at: https://www.cog-genomics.org/plink/2.0/.

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

## Acknowledgements

We thank the research participants and employees of 23andMe for making this work possible. We are grateful to S. Sunyaev, S. Chun, L. O'Connor, O. Weissbrod, and H. Finucane for helpful discussions. S.S.K. was supported by NIH award F31HG010818. This research was conducted using the UK Biobank Resource under Application #16549 and was funded by NIH grants R01 GM105857, R01 MH101244, U01 HG009379, and R01 HG006399..

## Author contributions

C.M.L. and A.L.P. designed experiments. C.M.L. performed experiments. C.M.L., S.G., P.R.L., S.S.K., N.F. and A.A. analyzed data. C.M.L. and A.L.P. wrote the paper with assistance from S.G., P.R.L. S.S.K., N.F. and A.L.P.

## Competing interests

The authors C.M.L., S.G., P.R.L., S.S.K. and A.L.P. declare no competing interests. N.F. and A.A. and members of the 23andMe research team are employees of 23andMe Inc.

## Additional information

## 23andMe Research Team

Michelle Agee[7], Babak Alipanahi[7], Robert K. Bell[7], Katarzyna Bryc[7], Sarah L. Elson[7], Pierre Fontanillas[7], David A. Hinds[7], Jey C. McCreight[7], Karen E. Huber[7], Aaron Kleinman[7], Nadia K. Litterman[7], Matthew H. McIntyre[7], Joanna L. Mountain[7], Elizabeth S. Noblin[7], Carrie A. M. Northover[7], Steven J. Pitts[7], J. Fah Sathirapongsasuti[7], Olga V. Sazonova[7], Janie F. Shelton[7], Suyash Shringarpure[7], Chao Tian[7], Joyce Y. Tung[7], Vladimir Vacic[7] & Catherine H. Wilson[7]

