## [Peer Review File · Nature Communications]

Reviewers' comments:

Reviewer #1 (Remarks to the Author):

This paper extends the LDpred model to LDpred-funct by weighting SNP effect sizes differently depending upon functional annotation information. The aim is to improve prediction accuracy by using prior biological knowledge. The claim is that using genomic annotation information improves out of sample prediction.

My main concern is that the manuscript contains a fundamental flaw. According to the authors, improvements in prediction accuracy are a function of the baseline-LD model v1.1 (75 functional annotation enrichments estimated using stratified LDSC). What is unclear is: how is the improvement specific to these functional annotations? An alternative hypothesis is that LDpred-funct performs better simply because rather than modelling SNP effects as coming from a 'spike and slab', they are modelled as coming from 75 different distributions each with their own weighting.

The improvements presented are relative to LD-pred. How does LD-pred-function compare to other summary statistics methods: SBLUP, SBayesR and RSS:

a. <https://www.biorxiv.org/content/biorxiv/early/2019/01/28/522961.full.pdf>

b. <https://www.ncbi.nlm.nih.gov/pmc/articles/PMC5796536/>

c. Robinson et al 2017 Nature Human Behaviour, Maier et al. 2018 Nature Communications, which has a slightly different weighting scheme (I'm not suggesting multi-trait, rather the shrinkage lambda values are different.)

The first two of these approaches fit multiple SNP effect size distributions. None of these four papers are referenced when they should be and it is imperative that comparisons are made with these approaches.

As a result, I feel there is not enough evidence that the improvement in prediction accuracy is specific to the biological information added to the model (which is the whole premise of the paper!).

Rather, would they get the same result if they had 25 mixture groups in summary-statistic bayesR, or even better, a shrinkage parameter per SNP as in RSS?

Additionally, the title is misleading as specific annotations from the baseline-LD model v1.1 are used and there is no presentation of how the model performs under different annotation models. To make a general statement, the authors would need to go much further, exploring different annotation models than just that used in the baseline LDscore approach.

Minor:

I think the cross-validation and how the annotations are used could be better explained. It's a bit confusing because I guess SNPs from every annotation should fall into each bin? For the cross-validation to work?

Reviewer #2 (Remarks to the Author):

Marquez-Luna et al present LD-Pred-Funct, a summary-statistics prediction method that incorporates functional information as a prior on effect sizes. They demonstrate on 21 UK Biobank traits, that LD-Pred-Funct outperforms LD-Pred (a quasi-precursor to LD-Pred-Funct), P+T (the classical / most-common method) and P+T+Lasso, a rival method.

Prediction is a very important area, with much effort being placed on achieving higher accuracy. In my view, while this paper has many strong points, it also has many deficiencies, both major and minor. However, I believe there is a high chance that all of these can be overcome, resulting in an excellent paper, and that LD-Pred-Funct has the potential to be a much-used method.

Please note, I am not familiar with AnnoPred, so am unable to comment whether LD-Pred-Funct is substantially different (the Discussion indicates the two methods certainly have some similarities).

Overall point

1 - My feeling is that this paper confuses two separate points:

A - you have a new method (while in theory an extension of LD-Pred, the changes are substantial - as well as allowing snp-specific priors, you have optimized the algorithm, I believe you no longer must divide the Genome into ~3000 segments, and while I do not agree with your tuning of parameters, this is a big change)

B - you are incorporating prior information from the S-LDSC model

Therefore, I believe you should have two separate analyses. The first would show your method is better, or as good as existing methods (ie show how it compares when run with the constant prior, where each snp expected to contribute equally). The second would show that the S-LDSC-derived priors are effective (i.e., show that accuracy improves when you switch from constant prior to per-SNP prior).

Instead, your current analysis mainly shows that the total impact of these two changes is to improve prediction, but it is not clear whether both changes are beneficial individually - I consider this important, as if only the first change improves prediction, then the main point of your paper should be that you have a new algorithm; while if only the second change improves prediction, the main point is the for prediction methods that allow for prior information, it is good to use priors derived from S-LDSC.

Major comments

2 - Tuning parameters using validation data. I am not happy with your suggested approach for using validation data to fit alpha parameters. This results in an unfair comparison, as most methods train the model using "365k" individuals, whereas LD-Pred-Funct uses 365k plus 90% of the 22k validation samples. To justify, you seem to cite papers that show this has been done for P+T, but that is usually a different scale (e.g., picking between maybe 5-8 p-value thresholds, whereas LD-Pred uses the validation samples to fit tens of parameters); further when papers report the best P+T model (across different P-value thresholds / clumping r^2), this is normally for convenience, and because if a paper is showing that a new method is better than P+T, then it is ok if P+T has an unfair advantage (but loses none-the-less). Ideally, you would have a way to picking the alphas from the training data (ie within the analysis that estimates effect sizes up to a constant); else, to ensure a fair comparison,

you should be performing the cross validation on the training data (exclude maybe 10% of the 350k individuals)

3 - Negative estimated h^2_j . From experience, the baseline ld model results in very many negative expected per-snp heritabilities (as in over 20%). In your analysis, you replace negative h^2_j with zero. While this is totally understandable, I believe it is correct for you to investigate how much your method performs better because of the relative weights given to snps, and how much because it first discards a huge chunk of snps (ie, how would LD-Pred-Funct do if h^2_j was replaced by $\max(h^2_j, 0)$).

4 - Rival methods. It is good you considered LD-Pred, but personally, I had not heard of P+T-Lasso. I would prefer you included at least one more major method, from the last year or so (e.g., SBayesR, or RSS, or something), just to reassure readers that your predictions are in comparable with those of the leading summary-statistic prediction methods).

5 - I was surprised to see that LD-Pred-Funct-Inf was only slightly better than LD-Pred-Inf (well done for including these results). You suggest that to be beneficial to include the per-SNP heritabilities requires a sparse prediction model, but our work found there were benefits for infinitesimal models. You might well be correct, and my results wrong. But on the other hand, is it possible your findings are a consequence of the (relatively) high relatedness in your training dataset (350k brits will likely be fairly related)? For example, relatedness gives the impression of a "highly polygenic prediction model" (the relatedness causes long-range ld, which means that more snps tag causal variants) - so conversely, the LD-Pred-Funct-Inf model might be capturing a large amount of "relatedness effects" which I could imagine being independent of per-SNP heritabilities. But that is just a suggestion, and I could easily be wrong.

Major, but easily fixable comments

6 - I disagree with the discussion sentence "We emphasize that our study is, to our knowledge, the first study that combines binary and continuous-valued functional annotations to improve polygenic risk prediction using imputed variants."

I consider this neither true technically, nor in spirit. We (Reference 51) constructed "Bayesian Polygenic Risk Scores" that harnessed binary annotations (e.g., our GCTA-Enriched Model (Baseline)) or continuous annotations (e.g., our LDK-Model) or both (e.g., our LDK-Enriched Model) - and for all of these we applied to imputed variants. Further, I don't agree with you distinguishing your method as being suitable for imputed variables (rather than just genotyped); yes, it is, but it is generally possible to apply summary-statistics method to imputed variants (unlike individual-level data methods, which might not be able to handle very large numbers of SNPs).

7 - The 23&Me data does not seem to add much, but introduces confusion. It is a shame adding in 23&Me (using a fixed effect meta-analysis) does not improve prediction, however, I can envisage why that might be (and you give some good reasons why). Instead, you only get prediction improvement when you introduce the two study weights (which does not really have anything to do with your new method, or the use of per-SNP heritabilities). However, I understand there are political reasons you might be required to keep in 23&Me data.

8 - The figures are poor quality - for 2 and 3, the trait names take up too much space, causing the bars to be very squeezed.

Minor comments

9 - Excellent number of secondary analyses - well done for care

10- Do you think it is necessary to mention UKBB and 23&Me in title? This gives the impression your method is only good for these datasets...). But this is only my view.

Signed Doug Speed

Reviewer #3 (Remarks to the Author):

This paper presents a new method for leveraging functional annotations when building polygenic prediction models from summary statistics. The method builds on the LDpred method to integrate variant specific functional annotation via the author's "baseline-LD" method. The baseline-LD method models functional annotations to provide SNP-specific heritability estimates, which in turn can be used as prior variances in the LDpred framework. This allows inference of posterior SNP weights that reflect both annotation information as well as the strength of associations in the trait's GWAS data. The method requires the optimisation of a number of tuning parameters --- in addition to LDpred's sparsity proportion, weighting parameters for each functional annotation SNP bin must

be fitted — which is done using the validation data. The authors argue that since the space of these tuning parameters is finite and relatively small compared to the number of samples in the validation data, the impact on over-fitting should be negligible.

The authors compare their method against a range of alternative approaches including a “vanilla” version of LDPred without functional annotations, as well as various simpler approaches such as p-value thresholding. The comparisons are made through application to 21 traits in the UK Biobank, as well as to height in a meta-analysis of UK Biobank and 23 and Me data (all under cross-validation). They also present results from a simulation study based on UK Biobank genotypes. They found consistent statistically significant improvements in predictions when incorporating functional annotations into predictions.

I enjoyed reading this paper. I think the method seems sound, pragmatically designed, and evidently scalable to summary statistics from large numbers of variants. Everything is well written and clearly explained, and caveats to interpretation and inference are carefully considered and explained at all stages. Clearly an enormous amount of work has been undertaken to produce this paper, both in the development of the framework and in the applications using 21 traits in the UK Biobank. While the predictive improvements from incorporating functional annotations demonstrated by the authors is not enormous, it is not negligible either, and even small predictive improvements could have important real world impact if polygenic scores are taken up more widely in healthcare (i.e. for disease screening etc). My comments are below:

Main Comments

1) I was rather surprised that the clearest comparator method “AnnoPred”, a polygenic prediction method which also incorporates functional annotation, was not mentioned in the introduction, and furthermore was apparently compared against but these comparisons were not included in the results section. Instead the comparisons were only briefly summarised in the discussion. Did I interpret correctly that the author’s method did not perform statistically significantly better than AnnoPred? I appreciate the distinction the authors draw to their own work but it is still a clear alternative method, and I think it should be included among the list of comparators throughout the results section, including all the figures — especially since the work in running it has apparently already been done. Also, given the lack of (if I understand correctly) statistically significant improvement offered by the author’s own method more of a detailed discussion of the relative merits of the two methods is warranted. How do the runtimes compare? Is their method easier to implement? Does their method offer any interpretation advantages?

2) Following on from the point above, the introduction is very brief, and the authors make little to no mention of the growing number of summary statistics polygenic risk methods that have emerged since LDPred. For example:

- Mak et al “Polygenic scores via penalized regression on summary statistics.” 2017 Gen Epi

- Newcombe et al "A flexible and parallelizable approach to genome-wide polygenic risk scores" 2019 Gen Epi

- Choi and O'Reilly "PRSice-2 Polygenic Risk Score software for biobank-scale data" 2019 GigaScience

In particular Mak et al's LassoSum has been around for a while, and in both Mak et al and Newcombe et al was found to offer more robust predictive performance than LDPred. Newcombe et al's JAMPred was found to perform similarly to LassoSum but can be faster (if the work is put in to set up parallelised analyses) and has the appealing feature of accounting for long range LD. Choi's PR-Sice 2 offers extremely fast computation for Biobank scale data. I would encourage the authors to read at least the introductions of these papers incase there are more summary statistic polygenic prediction methods to mention too. After reflection I do not think it is essential the authors include comparisons to any more of LDPred's alternatives since I take the main point as comparing the same method with and without functional annotation information, and showing the improvement therein. However, this is therefore more of a generic finding; a perspective I would encourage the authors to highlight better in the discussion. Providing more of a literature summary of other summary statistics predictive methods in the introduction would help make this point.

3) If I followed correctly all simulations were based on a genetic heritability similar to height (h_g^2 is set to 0.5), one of the most heritable complex traits. I suggest expanding the simulation space to include a lower heritability trait too, perhaps reflecting the range heritabilities across the 21 traits analysed in the UK Biobank.

4) Related to the point above, it worthwhile including a discussion/comment somewhere on the relationship between heritability and how much improvement incorporating functional annotation makes. Do the authors have any general perspectives to share after working on 21 traits? Is the improvement from functional annotation diluted in less heritable traits or do they feel the relative improvement is unaffected?

5) Perhaps I missed it but I didn't see any discussion of computational times. For example, how much longer does it take to run LDPred-funct compared to LDPred? I.e. accounting for not only the additional step in the method involving baseline-LD, but also the wider space of tuning parameters to explore.

Minor comments

1) I believe equation (6) should be a "max" function not a "min" function? On the previous page it is stated that K is between 1 and 100.

Response to reviewers for NCOMMS-19-24981 (Marquez-Luna et al.)

Reviewer #1 (Numbers added to reviewer comments):

This paper extends the LDpred model to LDpred-funct by weighting SNP effect sizes differently depending upon functional annotation information. The aim is to improve prediction accuracy by using prior biological knowledge. The claim is that using genomic annotation information improves out of sample prediction.

We thank the reviewer for the accurate summary of the manuscript.

1. My main concern is that the manuscript contains a fundamental flaw. According to the authors, improvements in prediction accuracy are a function of the baseline-LD model v1.1 (75 functional annotation enrichments estimated using stratified LDSC). What is unclear is: how is the improvement specific to these functional annotations? An alternative hypothesis is that LDpred-funct performs better simply because rather than modelling SNP effects as coming from a 'spike and slab', they are modelled as coming from 75 different distributions each with their own weighting.

The reviewer has raised the question of whether the improvement is specific to the 75 functional annotations of the baseline-LD model used by LDpred-funct. To address this, we implemented an analogous method that uses 75 random annotations: LDpred-funct (random). We also implemented a method that assumes the same prior distribution for all variants and then uses cross-validation to regularize these estimates (equivalent to LDpred-funct with no functional annotations): LDpred-inf + sparsity. (Also see response to Reviewer #2 Comment 1).

Across 21 UK Biobank traits, we determined that LDpred-funct attains a +13% relative improvement in average prediction \$R^2\$ vs. LDpred-funct (random), which performs similarly to LDpred (3.1% decrease in average prediction \$R^2\$ ), and that LDpred-funct attains a +13% relative improvement in average prediction \$R^2\$ vs. LDpred-inf + sparsity. Thus, the improvement of LDpred-funct is specific to the 75 functional annotations of the baseline-LD model used by LDpred-funct. We have updated the Application to 21 UK Biobank traits subsection of the Results section (p.16), citing Table S17, to include these results. We thank the reviewer for suggesting these experiments, and believe that the manuscript is improved as a result.

2. The improvements presented are relative to LD-pred. How does LD-pred-function compare to other summary statistics methods: SBLUP, SBayesR and RSS:

a. <https://www.biorxiv.org/content/biorxiv/early/2019/01/28/522961.full.pdf>

b. <https://www.ncbi.nlm.nih.gov/pmc/articles/PMC5796536/>

c. Robinson et al 2017 Nature Human Behaviour, Maier et al. 2018 Nature Communications, which has a slightly different weighting scheme (I'm not suggesting multi-trait, rather the shrinkage lambda values are different.)

The first two of these approaches fit multiple SNP effect size distributions. None of these four papers are referenced when they should be and it is imperative that comparisons are made with these approaches.

As a result, I feel there is not enough evidence that the improvement in prediction accuracy is specific to the biological information added to the model (which is the whole premise of the paper!). Rather, would they get the same result if they had 25 mixture groups in summary-statistic bayesR, or even better, a shrinkage parameter per SNP as in RSS?

We agree that it is of interest to compare LDpred-funct to other summary statistic based methods, not just to P+T and LDpred and P+T-funct-LASSO and AnnoPred as in the previously submitted version of our manuscript. Of the 3 methods mentioned by the reviewer, we elected to focus on SBayerR (Lloyd-Jones et al. 2019 Nat Commun; ref. 9), because Lloyd-Jones et al. reported that SBayerR performed as well or better than both SBLUP and RSS (Figure 2 of ref. 9) and was more computationally efficient. We have thus updated the Methods section (p.5) to include a description of SBayerR. We have also updated the Introduction section (p.3) to cite RSS (Zhu & Stephens 2017 Ann Appl Stat; ref. 12), SBLUP (Maier et al. 2018 Nat Commun; ref. 11), and SBayerR (Lloyd-Jones et al. 2019 Nat Commun; ref. 9).

Across 21 UK Biobank traits, we determined that LDpred-funct attains a 4.6% relative improvement in average prediction R^2 vs. SBayerR ($P=0.04$ for difference using two-sided z-test based on block-jackknife standard error). We have updated the Application to 21 UK Biobank traits subsection of the Results section (p.15, updating Figure 2 and Figure 3, to include this result. We also ran simulations with SBayerR, but determined that SBayerR attained prediction R^2 close to 0 at larger sample sizes ($N=20K$ and $N=50K$), perhaps because the algorithm failed to converge (we observed an estimated $h^2g=0.99$ in 100/100 simulations). We have updated the Simulations subsection of the Results section (p.13), citing Table S3, to note this result. We observed a similar phenomenon in one of the analyses in the "Application to height in meta-analysis of UK Biobank and 23andMe cohorts" subsection of the Results section, and have updated this subsection (p.18), citing Table S26, to note this result. We thank the reviewer for suggesting the comparison to SBayerR, and believe that the manuscript is improved as a result. (Also see response to Reviewer #2 Comment 4 and response to Reviewer #3 Comment 2).

3. Additionally, the title is misleading as specific annotations from the baseline-LD model v1.1 are used and there is no presentation of how the model performs under different annotation models. To make a general statement, the authors would need to go much further, exploring different annotation models then just that used in the baseline LDscore approach.

The reviewer makes a good point that our previous title focused on the improvement provided by functional annotations, whereas the main purpose of the manuscript is to introduce a new

polygenic prediction method, LDpred-funct. We have thus changed the title to “LDpred-funct: incorporating functional priors improves polygenic prediction accuracy in UK Biobank and 23andMe data sets”, which we believe is a more methodologically focused title. We have updated the Abstract (p. 2) and Introduction (p.3) accordingly.

We believe that a comprehensive assessment of how much different functional annotation models contribute to improvements in prediction accuracy is outside the scope of this manuscript. However, we have updated the Discussion section (p.19) to note the importance of this future direction, particularly as functional annotation models will improve as increasingly rich functional data is generated.

Minor:

I think the cross-validation and how the annotations are used could be better explained. It's a bit confusing because I guess snps from every annotation should fall into each bin? For the cross-validation to work?

We recognize that it is our responsibility to provide a clear exposition. We believe that the reviewer is referring to the step of LDpred-funct that regularizes posterior mean causal effect sizes by partitioning the SNPs into K bins (with roughly the same sum of squared posterior mean effect sizes) and using cross-validation to re-weight posterior mean causal effect sizes in each bin. We note that functional annotations are not used in this step (although they do impact the posterior mean causal effect size provided as input to this step). The reviewer is correct that SNPs from a given functional annotation are expected to fall into different bins (possibly all of the bins). We have updated the Methods section (p.8) to clarify these points.

Reviewer #2 (Remarks to the Author):

Marquez-Luna et al present LD-Pred-Funct, a summary-statistics prediction method that incorporates functional information as a prior on effect sizes. They demonstrate on 21 UK Biobank traits, that LD-Pred-Funct outperforms LD-Pred (a quasi-precursor to LD-Pred-Funct), P+T (the classical / most-common method) and P+T+Lasso, a rival method.

Prediction is a very important area, with much effort being placed on achieving higher accuracy. In my view, while this paper has many strong points, it also has many deficiencies, both major and minor. However, I believe there is a high chance that all of these can be overcome, resulting in an excellent paper, and that LD-Pred-Funct has the potential to be a much-used method.

We thank the reviewer for the accurate summary of our work, and for suggesting that there is the potential for our paper to be excellent and for our method to be much-used.

Please note, I am not familiar with AnnoPred, so am unable to comment whether LD-Pred-Funct is substantially different (the Discussion indicates the two methods certainly have some similarities).

We have made 5 changes to our presentation of the previous AnnoPred method (Hu et al. 2017 PLoS Comput Biol; ref. 24) (also see response to Reviewer #3 Comment 1):

First, we now explicitly mention AnnoPred in the Introduction section (p.3)

Second, we have added a Methods subsection summarizing the AnnoPred method (p. 6).

Third, we have added simulations with AnnoPred to the Simulations subsection of the Results section (p.13), updating Figure 1. We determined that, LDpred-funct outperformed AnnoPred with statistically significant (up to $p < 10^{-125}$) difference for large sample sizes across different genetic architectures, while AnnoPred was moderately superior (up to $p < 10^{-3}$) for smaller samples sizes (see Table S4).

Fourth, we have moved the analysis of real traits with AnnoPred to the Application to 21 UK Biobank traits subsection of the Results section (p.15), updating Figure 2 and Figure 3. As in the previous version of our manuscript, we determined that AnnoPred performed slightly but non-significantly worse than LDpred-funct (-2.7 % relative change in average prediction R^2 for AnnoPred vs. LDpred-funct see Table S17, $P=0.35$ for difference using two-sided z-test based on block-jackknife standard error in Table S16). We have also updated Figure 4 (cited on p.17) in the “Application to height in meta-analysis of UK Biobank and 23andMe cohorts” subsection of the Results section.

Fifth, we have updated the Discussion section (p.18) to further discuss the differences between AnnoPred and LD-pred-funct.

Overall point

1 - My feeling is that this paper confuses two separate points:

A - you have a new method (while in theory an extension of LD-Pred, the changes are substantial - as well as allowing snp-specific priors, you have optimized the algorithm, I believe you no longer must divide the Genome into ~3000 segments, and while I do not agree with your tuning of parameters, this is a big change)

We agree with the reviewer that an alternative possibility is to use banding (Yang et al. 2012 Nat Genet; ref. 34), and have updated the Methods section (p.7) to note this point.

B - you are incorporating prior information from the S-LDSC model

The reviewer is correct that we are incorporating prior information from the baseline-LD model used by S-LDSC.

Therefore, I believe you should have two separate analyses. The first would show your method is better, or as good as existing methods (ie show how it compares when run with the constant prior, where each snp expected to contribute equally). The second would show that the S-LDSC-derived priors are effective (i.e., show that accuracy improves when you switch from constant prior to per-SNP prior).

Instead, your current analysis mainly shows that the total impact of these two changes is to improve prediction, but it is not clear whether both changes are beneficial individually - I consider this important, as if only the first change improves prediction, then the main point of your paper should be that you have a new algorithm; while if only the second change improves prediction, the main point is the for prediction methods that allow for prior information, it is good to use priors derived from S-LDSC.

The reviewer has raised the question of whether the improvement is specific to the 75 functional annotations of the baseline-LD model used by LDpred-funct. To address this, we implemented an analogous method that uses 75 random annotations: LDpred-funct (random). We also implemented a method that assumes the same prior distribution for all variants and then uses cross-validation to regularize these estimates (equivalent to LDpred-funct with no functional annotations): LDpred-inf + sparsity. (Also see response to Reviewer #1 Comment 1).

Across 21 UK Biobank traits, we determined that LDpred-funct attains a +13% relative improvement in average prediction R^2 vs. LDpred-funct (random), which performs similarly to LDpred (3.1% decrease in average prediction R^2), and that LDpred-funct attains a +13% relative improvement in average prediction R^2 vs. LDpred-inf + sparsity. Thus, the improvement of LDpred-funct is specific to the 75 functional annotations of the baseline-LD model used by LDpred-funct. We have updated the Application to 21 UK Biobank traits subsection of the Results section (p.16), citing Table S17, to include these results. We thank the reviewer for suggesting these experiments, and believe that the manuscript is improved as a result.

Major comments

2 - Tuning parameters using validation data. I am not happy with your suggested approach for using validation data to fit alpha parameters. This results in an unfair comparison, as most methods train the model using "373k" individuals, whereas LD-Pred-Funct uses 373k plus 90% of the 22k validation samples. To justify, you seem to cite papers that show this has been done for P+T, but that is usually a different scale (e.g., picking between maybe 5-8 p-value thresholds, whereas LD-Pred uses the validation samples to fit tens of parameters); further when papers report the best P+T model (across different P-value thresholds / clumping r^2), this is normally for convenience, and because if a paper is showing that a new method is better than P+T, then it is ok if P+T has an unfair advantage (but loses none-the-less). Ideally, you would have a way to picking the alphas from the training data (ie within the analysis that estimates effect sizes up to a constant); else, to ensure a fair comparison, you should be performing the cross validation on the training data (exclude maybe 10% of the 350k individuals).

The reviewer is correct that LDpred-funct used 373K samples for training plus an additional 90% of the 22K validation samples for tuning alpha parameters for the sparsity step. We agree with the reviewer that it is important to confirm that this does not result in an unfair advantage to LDpred-funct relative to the other methods compared to. We further agree that this confirmation should be provided empirically, and not just by drawing an analogue to P+T.

We assessed this in two ways. First, we assessed the sensitivity of LDpred-funct to validation sample size. We determined that results were little changed when restricting to smaller validation sample sizes (as low as 1,000; see Table S22); in fact, average R^2 actually *increased* by 0.8%. Second, we implemented a version of LDpred-funct that uses only 1K samples for tuning alpha parameters for the sparsity step, applied to all 22K validation samples. We determined that results were virtually unchanged. Given that the difference between 373K training samples vs. 373K+1K=374K training samples is a miniscule relative difference, we believe that this empirically confirms that the use of additional validation samples for tuning does not result in an unfair advantage to LDpred-funct. We have updated the Application to 21 UK Biobank traits subsection of the Results section (p.15) to include a separate paragraph carefully discussing this point, citing Table S23. We have conservatively elected to retain the analysis using 90% of the 22K validation samples for tuning alpha parameters as the primary analysis (in preference to the analysis using only 1K samples for tuning alpha parameters, which further increased average R^2 by 0.8%; we do not wish to over fit the data and overstate our results). However, if the reviewer has a strong preference, we are willing to report the analysis using only 1K samples for tuning alpha parameters as the primary analysis, in order to maximally emphasize that there is no unfair advantage to LDpred-funct from using additional validation samples. We have also updated our discussion of this point in the Discussion section (p.19).

3 - Negative estimated h^2_j . From experience, the baseline ld model results in very many negative expected per-snp heritabilities (as in over 20%). In your analysis, you replace negative h^2_j with zero. While this is totally understandable, I believe it is correct for you to investigate how much your method performs better because of the relative weights given to snps, and how much because it first discards a huge chunk of snps (ie, how would LD-Pred-Funct do if h^2_j was replaced by $\max(h^2_j, 0)$).

We agree that is of interest to assess how much of the improvement LDpred-funct derives from the removal of (relatively) uninformative SNPs. To assess this, we implemented an analogous method that restricts the SNPs set in the same way but then uses a constant prior: LDpred-funct (constant prior).

Across 21 UK Biobank traits, we determined that LDpred-funct attains a 4.3% relative improvement in average prediction R^2 vs. LDpred-funct (constant prior), which attains a 5.5% relative improvement in average prediction R^2 vs. LDpred. Thus, some but not all of the improvement of LDpred-funct derives from the removal of (relatively) uninformative SNPs. We have updated the Application to 21 UK Biobank traits subsection of the Results section (p.16-

17), citing Table S17, to include this result. We thank the reviewer for suggesting this experiment, and believe that the manuscript is improved as a result.

4 - Rival methods. It is good you considered LD-Pred, but personally, I had not heard of P+T-Lasso. I would prefer you included at least one more major method, from the last year or so (e.g., SBayesR, or RSS, or something), just to reassure readers that your predictions are in comparable with those of the leading summary-statistic prediction methods).

We agree that it is of interest to compare LDpred-funct to other summary statistic based methods, not just to P+T and LDpred and P+T-funct-LASSO and AnnoPred as in the previously submitted version of our manuscript. Of the 3 methods mentioned by the reviewer, we elected to focus on SBayesR (Lloyd-Jones et al. 2019 Nat Commun; ref. 9), because Lloyd-Jones et al. reported that SBayesR performed as well or better than both SBLUP and RSS (Figure 2 of ref. 9) and was more computationally efficient. We have thus updated the Methods section (p.5) to include a description of SBayesR. We have also updated the Introduction section (p.3) to cite RSS (Zhu & Stephens 2017 Ann Appl Stat; ref. 12), SBLUP (Maier et al. 2018 Nat Commun; ref. 11), and SBayesR (Lloyd-Jones et al. 2019 Nat Commun; ref. 9).

Across 21 UK Biobank traits, we determined that LDpred-funct attains a 4.6% relative improvement in average prediction R^2 vs. SBayesR ($P=0.04$ for difference using two-sided z-test based on block-jackknife standard error). We have updated the Application to 21 UK Biobank traits subsection of the Results section (p.15, updating Figure 2 and Figure 3, to include this result. We also ran simulations with SBayesR, but determined that SBayesR attained prediction R^2 close to 0 at larger sample sizes ($N=20K$ and $N=50K$), perhaps because the algorithm failed to converge (we observed an estimated $h^2g=0.99$ in 100/100 simulations). We have updated the Simulations subsection of the Results section (p.13), citing Table S3, to note this result. We observed a similar phenomenon in one of the analyses in the "Application to height in meta-analysis of UK Biobank and 23andMe cohorts" subsection of the Results section, and have updated this subsection (p.18), citing Table S26, to note this result. We thank the reviewer for suggesting the comparison to SBayesR, and believe that the manuscript is improved as a result. (Also see response to Reviewer #1 Comment 2 and response to Reviewer #3 Comment 2).

5 - I was surprised to see that LD-Pred-Funct-Inf was only slightly better than LD-Pred-Inf (well done for including these results). You suggest that to be beneficial to include the per-SNP heritabilities requires a sparse prediction model, but our work found there were benefits for infinitesimal models. You might well be correct, and my results wrong. But on the other hand, is it possible your findings are a consequence of the (relatively) high relatedness in your training dataset (350k SNPs will likely be fairly related)? For example, relatedness gives the impression of a "highly polygenic prediction model" (the relatedness causes long-range LD, which means that more SNPs tag causal variants) - so conversely, the LD-Pred-Funct-Inf model might be capturing a large amount of "relatedness effects" which I could imagine being independent of per-SNP heritabilities. But that is just a suggestion, and I could easily be wrong.

Across 21 UK Biobank traits, the average prediction R^2 was 0.1126 for LDpred-inf and 0.1343 for LDpred-funct-inf, an improvement of 19% vs. LDpred-inf ($P < 10^{-20}$ for difference) (Table S20). We recognize that it is our responsibility to provide a clear exposition. We have updated the Application to 21 UK Biobank traits subsection of the Results section (p.15), citing Table S20, to clarify that LDpred-funct-inf substantially outperforms LDpred-inf.

Major, but easily fixable comments

6 - I disagree with the discussion sentence "We emphasize that our study is, to our knowledge, the first study that combines binary and continuous-valued functional annotations to improve polygenic risk prediction using imputed variants."

I consider this neither true technically, nor in spirit. We (Reference 51) constructed "Bayesian Polygenic Risk Scores" that harnessed binary annotations (e.g., our GCTA-Enriched Model (Baseline)) or continuous annotations (e.g., our LDAK-Model) or both (e.g., our LDAK-Enriched Model) - and for all of these we applied to imputed variants. Further, I don't agree with you distinguishing your method as being suitable for imputed variables (rather than just genotyped); yes, it is, but it is generally possible to apply summary-statistics method to imputed variants (unlike individual-level data methods, which might not be able to handle very large numbers of SNPs).

We have deleted this sentence from the Discussion section.

7 - The 23&Me data does not seem to add much, but introduces confusion. It is a shame adding in 23&Me (using a fixed effect meta-analysis) does not improve prediction, however, I can envisage why that might be (and you give some good reasons why). Instead, you only get prediction improvement when you introduce the two study weights (which does not really have anything to do with your new method, or the use of per-SNP heritabilities). However, I understand there are political reasons you might be required to keep in 23&Me data.

We were initially surprised that adding in 23&Me (increasing the training samples size from 408K to 1107K) did not improve substantially improve prediction accuracy. (We now believe that this is a logical result, due to the higher heritability in UK Biobank.) We have elected to retain this subsection in the main text, both because it provides an independent validation in a different data set of the *relative* improvement attained by LDpred-funct, and because we believe that the absence of a substantial improvement in *absolute* prediction accuracy (independent of which method is used) will be of interest to readers. However, we are willing to move the "Application to height in meta-analysis of UK Biobank and 23andMe cohorts" subsection to a Supplementary Note if the reviewer and/or editors have a strong preference.

8 - The figures are poor quality - for 2 and 3, the trait names take up too much space, causing the bars to be very squeezed.

We have updated Figure 2 and Figure 3 by reducing the amount of space allocated to the trait names.

Minor comments

9 - Excellent number of secondary analyses - well done for care

We thank the reviewer for acknowledging the large number of secondary analyses in our previously submitted manuscript. We have added many new analyses to the current version.

10- Do you think it is necessary to mention UKBB and 23&Me in title? This gives the impression your method is only good for these datasets...). But this is only my view.

We prefer to include the UKBB and 23&Me data sets in the title in order to make clear that the study is impactful not only because it develops a new statistical method, but also in its application of the method to large empirical data sets. We have elected to retain the names of these data sets in the title, but are willing to remove them if the reviewer and/or editors have a strong preference.

Signed Doug Speed

Reviewer #3 (Remarks to the Author):

This paper presents a new method for leveraging functional annotations when building polygenic prediction models from summary statistics. The method builds on the LDPred method to integrate variant specific functional annotation via the author's "baseline-LD" method. The baseline-LD method models functional annotations to provide SNP-specific heritability estimates, which in turn can be used as prior variances in the LDPred framework. This allows inference of posterior SNP weights that reflect both annotation information as well as the strength of associations in the trait's GWAS data. The method requires the optimisation of a number of tuning parameters --- in addition to LDPred's sparsity proportion, weighting parameters for each functional annotation SNP bin must be fitted --- which is done using the validation data. The authors argue that since the space of these tuning parameters is finite and relatively small compared to the number of samples in the validation data, the impact on over-fitting should be negligible.

The authors compare their method against a range of alternative approaches including a "vanilla" version of LDPred without functional annotations, as well as various simpler approaches such as p-value thresholding. The comparisons are made through application to 21 traits in the UK Biobank, as well as to height in a meta-analysis of UK Biobank and 23 and Me data (all under cross-validation). They also present results from a simulation study based on UK Biobank genotypes. They found consistent statistically significant improvements in predictions when incorporating functional annotations into predictions.

I enjoyed reading this paper. I think the method seems sound, pragmatically designed, and evidently scalable to summary statistics from large numbers of variants. Everything is well

written and clearly explained, and caveats to interpretation and inference are carefully considered and explained at all stages. Clearly an enormous amount of work has been undertaken to produce this paper, both in the development of the framework and in the applications using 21 traits in the UK Biobank. While the predictive improvements from incorporating functional annotations demonstrated by the authors is not enormous, it is not negligible either, and even small predictive improvements could have important real world impact if polygenic scores are taken up more widely in healthcare (i.e. for disease screening etc). My comments are below:

We thank the reviewer for the accurate summary of our work, for suggesting that the method is sound, pragmatically designed, and scalable, and for acknowledging the enormous amount of work that has been undertaken. We have added many new analyses to the current version.

Regarding “the impact of over-fitting should be negligible”: see response to Reviewer #2 Comment 2 on tuning parameters and concerns about over-fitting.

Main Comments

1) I was rather surprised that the clearest comparator method “AnnoPred”, a polygenic prediction method which also incorporates functional annotation, was not mentioned in the introduction, and furthermore was apparently compared against but these comparisons were not included in the results section. Instead the comparisons were only briefly summarised in the discussion. Did I interpret correctly that the author’s method did not perform statistically significantly better than AnnoPred? I appreciate the distinction the authors draw to their own work but it is still a clear alternative method, and I think it should be included among the list of comparators throughout the results section, including all the figures — especially since the work in running it has apparently already been done. Also, given the lack of (if I understand correctly) statistically significant improvement offered by the author’s own method more of a detailed discussion of the relative merits of the two methods is warranted. How do the runtimes compare? Is their method easier to implement? Does their method offer any interpretation advantages?

We have made 5 changes to our presentation of the previous AnnoPred method (Hu et al. 2017 PLoS Comput Biol; ref. 24) (also see response to Reviewer #2 Introductory comments, paragraph 3)

First, we now explicitly mention AnnoPred in the Introduction section (p.3)

Second, we have added a Methods subsection summarizing the AnnoPred method (p. 6).

Third, we have added simulations with AnnoPred to the Simulations subsection of the Results section (p.13), updating Figure 1. We determined that, LDpred-funct outperformed AnnoPred with statistically significant (up to $p < 10^{-125}$) difference for large sample sizes across different genetic architectures, while AnnoPred was moderately superior (up to $p < 10^{-3}$) for smaller samples sizes (see Table S4).

Fourth, we have moved the analysis of real traits with AnnoPred to the Application to 21 UK Biobank traits subsection of the Results section (p.15), updating Figure 2 and Figure 3. As in the previous version of our manuscript, we determined that AnnoPred performed slightly but non-significantly worse than LDpred-funct (-2.7 % relative change in average prediction R^2 for AnnoPred vs. LDpred-funct see Table S17, $P=0.35$ for difference using two-sided z-test based on block-jackknife standard error in Table S16). We have also updated Figure 4 (cited on p.17) in the “Application to height in meta-analysis of UK Biobank and 23andMe cohorts” subsection of the Results section.

Fifth, we have updated the Discussion section (p.18) to further discuss the differences between AnnoPred and LD-pred-funct.

2) Following on from the point above, the introduction is very brief, and the authors make little to no mention of the growing number of summary statistics polygenic risk methods that have emerged since LDPred. For example:

- Mak et al “Polygenic scores via penalized regression on summary statistics.” 2017 Gen Epi
- Newcombe et al “A flexible and parallelizable approach to genome-wide polygenic risk scores” 2019 Gen Epi
- Choi and O’Reilly “PRSice-2 Polygenic Risk Score software for biobank-scale data” 2019 GigaScience

In particular Mak et al’s LassoSum has been around for a while, and in both Mak et al and Newcombe et al was found to offer more robust predictive performance than LDPred. Newcombe et al’s JAMPred was found to perform similarly to LassoSum but can be faster (if the work is put in to set up parallelised analyses) and has the appealing feature of accounting for long range LD. Choi’s PR-Sice 2 offers extremely fast computation for Biobank scale data. I would encourage the authors to read at least the introductions of these papers incase there are more summary statistic polygenic prediction methods to mention too. After reflection I do not think it is essential the authors include comparisons to any more of LDPred’s alternatives since I take the main point as comparing the same method with and without functional annotation information, and showing the improvement there-in. However, this is therefore more of a generic finding; a perspective I would encourage the authors to highlight better in the discussion. Providing more of a literature summary of other summary statistics predictive methods in the introduction would help make this point.

We have updated the Introduction section (p.3) to cite all of the methods mentioned by the reviewer, as well as RSS (Zhu & Stephens 2017 Ann Appl Stat; ref. 12), SBLUP (Maier et al. 2018 Nat Commun; ref. 11), and SBayesR (Lloyd-Jones et al. 2019 Nat Commun; ref. 9) (also see response to Reviewer #1 Comment 2 and response to Reviewer #2 Comment 4).

Although the reviewer has indicated that it is not essential to include comparisons to any more of LDpred’s alternatives, we have elected (based on the feedback of Reviewer 1 and Reviewer 2) to add a comparison to SBayesR. See response to Reviewer #1 Comment 2 and response to Reviewer #2 Comment 4.

We have updated the Discussion section (p.18) to emphasize that the main finding is that incorporating functional annotations improves polygenic prediction accuracy. We note that many state-of-the-art methods (including SBayesR; Lloyd-Jones et al. 2019 Nat Commun) do not currently support incorporation of functional annotations.

3) If I followed correctly all simulations were based on a genetic heritability similar to height (h_g^2 is set to 0.5), one of the most heritable complex traits. I suggest expanding the simulation space to include a lower heritability trait too, perhaps reflecting the range heritabilities across the 21 traits analyzed in the UK Biobank.

We have added a new set of simulations with h_g^2 set to 0.25 instead of 0.5. We determined that the impact of reducing h_g^2 was generally similar to that of reducing training sample size (see Table S12). We have updated the Simulations subsection of the Results section (p.14), citing Table S12, to report these results.

4) Related to the point above, it worthwhile including a discussion/comment somewhere on the relationship between heritability and how much improvement incorporating functional annotation makes. Do the authors have any general perspectives to share after working on 21 traits? Is the improvement from functional annotation diluted in less heritable traits or do they feel the relative improvement is unaffected?

Distinct from our simulation analyses (see response to Reviewer #3 Comment 3), we have generated a plot of the relative improvement of LDpred-funct over LDpred for 21 UK Biobank traits vs. h_g^2 (measured on the observed scale for binary traits, and excluding two sex-specific traits). The plot is displayed in Figure S2, cited in the Application to 21 UK Biobank traits subsection of the Results section (p.15). We observed a positive but non-significant correlation across traits between h_g^2 and relative improvement, perhaps due to the limited number of data points and/or contribution of other factors (e.g. polygenicity).

5) Perhaps I missed it but I didn't see any discussion of computational times. For example, how much longer does it take to run LDpred-funct compared to LDpred? I.e. accounting for not only the additional step in the method involving baseline-LD, but also the wider space of tuning parameters to explore.

We agree that it is of interest to report running times. We now report running times (based on chr1 simulations, extrapolated to the entire genome) for the 7 main methods compared. Results are reported in Table S8, cited in the Simulations subsection of the Results section (p.13-14). We determined that LDpred-funct (71 min) is 73 times faster than AnnoPred (5,249 min), the second best method in terms of prediction R^2 in analyses of real UK Biobank traits.

Minor comments

1) I believe equation (6) should be a "max" function not a "min" function? On the previous page it is stated that K is between 1 and 100.

We believe that “min” in equation 6 (Methods section, p.8) is correct. This ensures that K ranges between 1 and 100, even for very large N .

Additional changes:

We have elected to perform a new analysis in which we evaluated the performance of LDpred-funct in predicting 21 UK Biobank traits in non-European populations (South Asians + Africans) using European training data (as in two recent studies motivating polygenic prediction in non-European populations as an important research priority: Martin et al. 2019 Nat Genet (ref. 49) and Duncan et al. 2019 Nat Commun (ref. 50)). Our results were promising, particularly in Africans (+23% vs. LDpred ($P < 10^{-5}$), +18% vs. SBayesR ($P = 0.001$); see Table S27), for which distinguishing causal vs. non-causal variants is particularly important due to differences in LD vs. Europeans⁵³. We believe that these findings substantially increase the impact of our work. We have updated the Discussion section (p.19), citing Table S27, to note these findings.

Also, we would like to disclose that due to a data issue the analyses in the previous version of the manuscript inadvertently included a small number related individuals in the validation data. We have updated all analyses to fix this issue. Our results are nearly unchanged, and our conclusions are unaffected.

Additional comment:

We note that when uploading the manuscript into the Submission System, the pdf file generated shifted the following page numbers:

Reviewer #2 Comment 2. P15 (in Word) changed to p16 (PDF generated by Submission system).

Reviewer #2 Comment 3. P16 (in Word) changed to p17 (PDF generated by Submission system).

Reviewer #3 Comment 5. P13 (in Word) changed to p14 (PDF generated by Submission system).

REVIEWER COMMENTS

Reviewer #1 (Remarks to the Author):

The authors have thoroughly addressed all of my concerns and presented their results clearly.

Reviewer #2 (Remarks to the Author):

In summary, the authors have carefully answered most of my points, but the one I am not happy with their response to, is the most important. I also have two new, minor comments

#####

Point 1 - I commented that (new) LDPredFunct has two innovations. First it introduces a new algorithm, then it incorporates priors. Therefore, I asked they run once with just new algorithm (and constant prior) and then with both.

They did this, thank you. For reference, I believe the two important comparisons are

$\text{LDPredFunct (constant prior)} > \text{LDPred-Inf}$

This shows that your new algorithm (ie binning SNPs) is better than not binning

and $\text{LDPredFunct} > \text{LDPredFunct (constant prior)}$

This shows that your new algorithm plus functions is better than your new algorithm without functions

It would also be interesting to mention that

$\text{LDPredFunct (constant prior)} > \text{LDPred}$

as this shows that your way of improving LDPred-inf (by binning) is better than improving LDPred-inf by MCMC

Ps, all the methods are confusing, and you understand them better than I, so please make sure what I said is sensible, and correct names if necessary!

Point 2 - I mentioned that you use validation data as training. I am very unhappy with your response

You have responded in two ways

First, you reduced the validation samples from 20k down to as low as 1k (then reporting accuracy for those subsets). I do not see much evidence from this analysis. You are continuing to use the validation samples for model fitting. If anything, it makes things worse, because reducing the size of the validation set might introduce more stochastic bias (by-chance r^2 from a linear regression of 1000 samples will be higher than by-chance r^2 from a linear regression of 20k samples, etc), and means you can not fairly compare results to those that measured accuracy using whole validation set. Moreover, I believe it is not fair to say 374k vs 373k is a "miniscule" difference, because the 373k and 1k samples contribute to analysis in very different ways; 373k contribute to training, whereas 1k contribute to testing.

Your second analysis used 1k validation samples to train, then tested over 22k samples. This is better, because you now only use 1k validation samples as training, not all 22k, so it is reassuring results are similar.

However, I do not see why you have not performed the analysis I suggested. I have now tried your software (it worked well, thanks). But I see it does not readily produce effect sizes. It produces them for the inf model, but not after the weighting. These would be required to do the analysis I suggest. Now I cant see how you performed the second analysis without these weights, so that suggests you are able to extract them.

In any case. It should be fairly simple to make the comparisons fair. All you need to do is repeat the single-snp analysis with say 5k less samples, then uses the held-out 5k as your validation set, get the effect sizes and project onto the original 22k validation samples.

In my view, this remains a major issue with the paper. Moreover, it should be relatively easy to fix (as in a couple of days worth of analysis, most of which time you can spend doing other things while the cluster runs).

Point 3 - thank you for thoroughly answering the question

Point 4 - thank you for finding a new method, I'm happy with this

Point 5 - I see I was wrong to say only a small difference. +19% is large

They also responded well to the remaining points

Additional comments

It is a big limitation your software does not report effect sizes for snps (for the final, bin-weighted model). This prevents you from transferring prs (effect sizes) between datasets

I believe caption to S23 is wrong (it seems to instead be caption to S22)

Doug Speed

Reviewer #3 (Remarks to the Author):

Thank you to the authors for their comprehensive response to all of the points I raised. These changes and additions, along with those made in response to the other reviewers, leave me with no further requests.

The paper is certainly stronger now, particularly for the new work that more clearly disentangles the benefit driven by functional annotation, and the new comparisons against another popular PRS method (SBayesR).

As indicated above, I have no further comments.

Response to reviewers for NCOMMS-19-24981 (Marquez-Luna et al.)

Reviewer #2 (Remarks to the Author):

In summary, the authors have carefully answered most of my points, but the one I am not happy with their response to, is the most important. I also have two new, minor comments

We thank the reviewer for indicating that most points have been carefully answered. The remaining comments are addressed below.

#####

Point 1 - I commented that (new) LDPredFunc has two innovations. First it introduces a new algorithm, then it incorporates priors. Therefore, I asked they run once with just new algorithm (and constant prior) and then with both.

They did this, thank you.

The reviewer is correct that we added a method called LDpred-funct(constant prior), as requested.

For reference, I believe the two important comparisons are

LDPredFunc (constant prior) > LDPred-Inf

This shows that your new algorithm (ie binning SNPs) is better than not binning

We agree that it is of interest to explicitly compare LDpred-funct(constant prior) to LDpred-inf. We determined that LDpred-funct (constant prior) attained a 23% relative improvement in average prediction R^2 vs. LDpred-inf, implying that regularizing causal effect size estimates in bins of different magnitude (i.e. binning SNPs, as implemented in LDpred-funct) is better than not binning SNPs; in addition, some of the improvement of LDpred-funct derives from the removal of (relatively) uninformative SNPs (10% relative improvement for LDpred-funct-inf (constant prior) vs. LD-pred-inf). We have updated the *Application to 21 UK Biobank traits* subsection of the Results section (p. 17, citing Table S17) to include this explicit comparison.

and LDPredFunc > LDPredFunc (constant prior)

This shows that your new algorithm plus functions is better than your new algorithm without functions

Indeed, we determined that LDpred-funct attained a 4.3% relative improvement in average prediction R^2 vs. LDpred-funct (constant prior), implying that including a prior informed by functional annotations is better than not including a prior informed by functional annotations. We have updated the *Application to 21 UK Biobank traits* subsection of the Results section (p. 17, citing Table S17) to explain this carefully.

It would also be interesting to mention that

LDPredFunc (constant prior) > LDPred

as this shows that your way of improving LDPred-inf (by binning) is better than improving LDPred-inf by MCMC

We determined that LDpred-funct (constant prior) attained a 5.5% relative improvement in average prediction R^2 vs. LDpred; this is noted in the the *Application to 21 UK Biobank traits* subsection of the Results section (p. 17, citing Table S17).

Ps, all the methods are confusing, and you understand them better than I, so please make sure what I said is sensible, and correct names if necessary!

Point 2 - I mentioned that you use validation data as training. I am very unhappy with your response

You have responded in two ways

First, you reduced the validation samples from 20k down to as low as 1k (then reporting accuracy for those subsets). I do not see much evidence from this analysis. You are continuing to use the validation samples for model fitting. If anything, it makes things worse, because reducing the size of the validation set might introduce more stochastic bias (by-chance r^2 from a linear regression of 1000 samples will be higher than by-chance r^2 from a linear regression of 20k samples, etc), and means you can not fairly compare results to those that measured accuracy using whole validation set. Moreover, I believe it is not fair to say 374k vs 373k is a "miniscule" difference, because the 373k and 1k samples contribute to analysis in very different ways; 373k contribute to training, whereas 1k contribute to testing.

Your second analysis used 1k validation samples to train, then tested over 22k samples. This is better, because you now only use 1k validation samples as training, not all 22k, so it is reassuring results are similar.

However, I do not see why you have not performed the analysis I suggested. I have now tried your software (it worked well, thanks). But I see it does not readily produce effect sizes. It produces them for the inf model, but not after the weighting. These would be required to do the analysis I suggest. Now I cant see how you performed the second analysis without these weights, so that suggests you are able to extract them.

In any case. It should be fairly simple to make the comparisons fair. All you need to do is repeat the single-snp analysis with say 5k less samples, then uses the held-out 5k as your validation set, get the effect sizes and project onto the original 22k validation samples.

We have performed the analysis requested by the reviewer. Specifically, we have:

- (i) Recomputed BOLT-LMM association statistics on up to 409K-5K=404K British samples, replacing the published BOLT-LMM statistics on up to 409K British samples. (We regret the delay in our resubmission; some technical issues arose in recomputing BOLT-LMM association statistics for some of the 21 traits.)
- (ii) Used (only) the 5K samples omitted from step (i) to estimate regularization weights.
- (iii) Computed predictions using the original set of up to 24K non-British European validation samples (average of 22K phenotyped non-British European validation samples).

We determined that results were little changed and not significantly different compared to our main analysis. In detail, we observed a 0.2% decrease in average prediction R^2 (from 0.1443 to 0.1440), but the decrease was not statistically significant ($P=0.98$). We have updated the *Application to 21 UK Biobank traits* subsection of the Results section (p. 15, citing Table S24) to include this new analysis.

Regarding the statement “You are continuing to use the validation samples for model fitting”: we would like to clarify that in every 10-fold cross-validation analysis in our paper, in each cross-validation fold, the validation samples used to estimate regularization weights are disjoint from the validation samples used to compute predictions. We have updated the LDpred-funct paragraph of the *Polygenic prediction methods* subsection of the Methods section (p.7-8) and the *Application to 21 UK Biobank traits* subsection of the Results section (p. 15) to clarify this point.

In my view, this remains a major issue with the paper. Moreover, it should be relatively easy to fix (as in a couple of days worth of analysis, most of which time you can spend doing other things while the cluster runs).

Point 3 - thank you for thoroughly answering the question

Point 4 - thank you for finding a new method, I'm happy with this

Point 5 - I see I was wrong to say only a small difference. +19% is large

We thank the reviewer for acknowledging our responses to these points.

They also responded well to the remaining points

We thank the reviewer for indicating that we responded well to the remaining points.

Additional comments

It is a big limitation your software does not report effect sizes for snps (for the final, bin-weighted model). This prevents you from transferring prs (effect sizes) between datasets

We have added a new flag to the software (--print-weights) that prints the regularized SNP effect sizes used for prediction. Our software is publicly available (see Web Resources).

I believe caption to S23 is wrong (it seems to instead be caption to S22)

We have corrected the caption for Table S23.

Doug Speed

REVIEWERS' COMMENTS

Reviewer #2 (Remarks to the Author):

Thanks for addressing my comments thoroughly, and I was pleased to see the additional analysis (sorry if I had misunderstood the cross-validation folds when I previously read the paper).

One last, very very minor comment. I would not describe RSS and SBLUP as "recently developed", as I believe they come from 2018 and 2017, respectively, (note that I think SBLUP was first described in <https://www.nature.com/articles/s41562-016-0016?proof=t>). Anyway that is just my suggestion, and thanks again for addressing my comments.

Doug

Response to reviewers for NCOMMS-19-24981 (Marquez-Luna et al.)

REVIEWERS' COMMENTS

Reviewer #2 (Remarks to the Author):

Thanks for addressing my comments thoroughly, and I was pleased to see the additional analysis (sorry if I had misunderstood the cross-validation folds when I previously read the paper).

One last, very very minor comment. I would not describe RSS and SBLUP as "recently developed", as I believe they come from 2018 and 2017, respectively, (note that I think SBLUP was first described in <https://www.nature.com/articles/s41562-016-0016?proof=t>). Anyway that is just my suggestion, and thanks again for addressing my comments.

Doug

We thank the reviewer for indicating that most points have been thoroughly addressed. The remaining comment has been addressed, and we removed the phrase "recently developed" when describing RSS and SBLUP.